# GABA, not BOLD, reveals dissociable learning-dependent plasticity mechanisms in the human brain

**Polytimi Frangou[1], Marta Correia[2], Zoe Kourtzi[1]\***

[1]Department of Psychology, University of Cambridge, Cambridge, United Kingdom; [2]MRC Cognition and Brain Sciences Unit, Cambridge, United Kingdom

**Abstract** Experience and training have been shown to facilitate our ability to extract and discriminate meaningful patterns from cluttered environments. Yet, the human brain mechanisms that mediate our ability to learn by suppressing noisy and irrelevant signals remain largely unknown. To test the role of suppression in perceptual learning, we combine fMRI with MR Spectroscopy measurements of GABA, as fMRI alone does not allow us to discern inhibitory vs. excitatory mechanisms. Our results demonstrate that task-dependent GABAergic inhibition relates to functional brain plasticity and behavioral improvement. Specifically, GABAergic inhibition in the occipito-temporal cortex relates to dissociable learning mechanisms: decreased GABA for noise filtering, while increased GABA for feature template retuning. Perturbing cortical excitability during training with tDCs alters performance in a task-specific manner, providing evidence for a direct link between suppression and behavioral improvement. Our findings propose dissociable GABAergic mechanisms that optimize our ability to make perceptual decisions through training.

DOI: https://doi.org/10.7554/eLife.35854.001

## Introduction

Understanding the structure of the world around us entails extracting and discriminating meaningful patterns from cluttered environments. Effortless as this may seem, it poses for the brain a challenging task that involves suppressing noisy and ambiguous sensory signals. Experience and training have been shown to facilitate perceptual judgments and visual recognition processes (*Fine and Jacobs, 2002*; *Gilbert et al., 2001*; *Goldstone, 1998*). For instance, an experienced bird watcher is not only able to break the camouflage and detect a bird in a leafy tree, but also determine whether it is a carrion crow or a hooded crow. Yet, the mechanisms that the human brain employs to suppress task-irrelevant information and optimize perceptual decisions through training remain largely unknown.

Theoretical models of perceptual learning (*Dosher et al., 2013*; *Dosher and Lu, 1998*; *Li et al., 2004*) posit that experience and training facilitate our ability to a) detect targets in clutter by filtering external noise, b) discriminate highly similar objects by suppressing irrelevant features and retuning task-relevant feature templates. Although considerable behavioral evidence supports this framework, its neural implementation remains uncertain. Previous fMRI studies have demonstrated changes in the overall activation of higher visual areas in the occipito-temporal cortex due to training on perceptual decision tasks (for reviews, see *Kourtzi, 2010*; *Welchman and Kourtzi, 2013*). However, fMRI data do not allow us to discern excitatory from suppressive mechanisms of experience-dependent plasticity, as BOLD reflects aggregate activity across large neural populations (*Heeger and Ress, 2002*; *Logothetis, 2008*).

Here, we ask whether GABA (γ-aminobutyric acid), the primary inhibitory neurotransmitter in the brain, mediates our ability to improve in making perceptual decisions through training. Previous

**Competing interests:** The authors declare that no competing interests exist.

**eLife digest** When searching for a friend in the crowd or telling identical twins apart, your visual system must solve a complex puzzle. It must ignore all irrelevant information (e.g., unfamiliar faces in the crowd) and focus on key features (e.g., your friend's familiar face) that will allow you to make a decision. We become better at solving complex visual discriminations with practice. But exactly how the brain achieves this improved performance is unclear.

To answer this question, Frangou et al. trained healthy volunteers on two such visual tasks. The first (target detection task) involved locating a target (e.g. circular shape made of dots among randomly distributed dots in the background), a task similar to identifying a friend in the crowd. The second (feature discrimination task) involved assigning highly alike shapes in two different categories, similar to telling apart identical twins. To solve this problem, volunteers had to identify distinct features that allowed them to distinguishthese shapes. During training on this task, they updated and refined the representation of these distinct features in their brain. This enabled them to make finer discriminations and assign each image correctly to one of the two categories.

While the volunteers trained on the tasks, Frangou et al. measured levels of a chemical called GABA in brain areas that process visual information. GABA is the brain's main inhibitory molecule and controls the activity of neurons. As the volunteers learned the two tasks, their brains showed opposite changes in GABA levels. In the first, target detection task, individuals did better if their GABA decreased during training. In the second, feature discrimination task, they achieved more if their GABA increased during training. To confirm these findings, Frangou et al. used a second technique to activate or suppress processing in visual areas of the brain. Activating visual areas enhanced performance on the target detection task. Suppressing them enhanced performance on the fine discrimination task. These changes are thus consistent with those seen in GABA levels.

As well as revealing how we learn to make decisions based on the information from our eyes, these findings suggest that adjusting brain activity could help patients regain skills lost as a result of eye-related or neurological conditions. Understanding the role of GABA in brain plasticity is also relevant to conditions like autism and psychosis, which have been shown to relate to changes in brain inhibition.

DOI: https://doi.org/10.7554/eLife.35854.002

work in animals has demonstrated that GABAergic inhibition is associated with learning and synaptic plasticity (*Castro-Alamancos et al., 1995*; *Trepel and Racine, 2000*). Yet, measuring GABA in the human brain has been possible only recently thanks to advances in MR Spectroscopy (MRS). Previous MRS studies have shown that GABA concentrations in the visual cortex relate to homeostatic plasticity (*Lunghi et al., 2015*), while GABA concentrations in the motor cortex relate to individual ability (*Kolasinski et al., 2017*; *Stagg et al., 2011a*) and improved performance (*Blicher et al., 2015*; *Floyer-Lea et al., 2006*; *O'Shea et al., 2017*; *Sampaio-Baptista et al., 2015*) in motor learning.

To probe the mechanisms that the human brain uses to suppress noisy and irrelevant signals, we employed two tasks that rely differentially on noise filtering vs. template retuning: (1) a signal-in-noise task that involves extracting a target masked by noise, (2) a feature-differences task that involves judging fine differences. Interrogating only fMRI signals does not allow us to discern between the brain mechanisms for noise filtering vs. template retuning: our results show similar learning dependent changes in behavioral performance and fMRI activation during training in both learning tasks. However, combining MRS measurements of GABA with fMRI uncovers distinct GABAergic inhibition mechanisms in the posterior occipito-temporal cortex. Learning to detect targets from clutter by noise filtering relates to decreased GABA, while learning to discriminate fine differences by template retuning relates to increased GABA. To move beyond correlative evidence, we then used transcranial direct current stimulation (tDCs) in the occipito-temporal cortex to perturb cortical excitability during training. Our findings relating GABAergic inhibition and behavioral improvement lead to opposite predictions for the effect of tDCs on the two learning tasks. In line with these predictions, we find dissociable effects of tDCs stimulation on behavioral improvement between tasks: excitatory anodal stimulation enhances learning to see in clutter, while inhibitory cathodal stimulation enhances learning feature differences. Thus, perturbing visual cortex

suppression alters behavioral improvement during training in a task-specific manner, providing evidence for a direct link between suppression and learning. Our findings propose that GABAergic inhibition in the visual cortex underlies dissociable learning mechanisms that optimize our ability to make perceptual decisions.

## Results

### Learning-dependent changes in behavior and fMRI

We tested two separate groups of participants on either (1) a signal-in-noise (SN) task that involves extracting shapes (radial vs. concentric Glass patterns) from background noise or (2) a feature-differences (FD) task that involves judging fine differences induced by morphing between the two stimulus classes (*Figure 1a*). For each task, participants improved during a single training session that took place during scanning (*Figure 1b*), consistent with previous reports showing fast behavioral improvement early in the training (for a review see *Sagi, 2011*). A repeated measures ANOVA (Task (SN vs. FD) x Training (training runs) showed significantly improved performance –as measured by d'– after training (main effect of Training: $F_{(6,192)}= 3.79$, p=0.001) but no significant effect of Task ($F_{(1,32)}=0.01$, p=0.91) nor Training x Task interaction ($F_{(6,192)}= 0.61$, p=0.722), suggesting similar improvement in both tasks. Testing participants the following day after training (transfer test) showed that performance was significantly different from the first training run for both tasks (main effect of Session: ($F_{(1,32)}= 10.59$, p=0.003; Task x Session interaction: ($F_{(1,32)}=0.89$, p=0.35) but not significantly different from the last training run (main effect of Session: $F_{(1,32)}=0.95$, p=0.34; Task x Session interaction: $F_{(1,32)}=0.18$, p=0.68), suggesting lasting performance improvement due to training. In contrast, no significant changes in performance were observed for a no-training control group who did not receive training in between test sessions (main effect of Session: $F_{(1,6)}= 1.13$, p=0.33; Task x Session interaction: $F_{(1,6)}=0.0003$, p=0.99) (*Figure 1—figure supplement 1*).

To further quantify behavioral improvement, we computed two complementary measures: a) delta d prime (Δd': last training run minus first training run) that indicates difference in perceptual sensitivity early vs. late in training, b) learning rate that indicates the rate with which perceptual sensitivity (d' calculated per training run) changes during training. These measures have been previously used in perceptual learning studies to quantify the effect of training on performance (*Ball and Sekuler, 1987*; *Chang et al., 2013*; *Dosher et al., 2013*). Behavioral improvement was similar between tasks, as indicated by no significant differences between tasks in learning rate (t(34)=0.03, p=0.974) nor Δd' (t(34)=0.806, p=0.426).

We next tested whether behavioral improvement relates to functional brain changes with learning. First, we tested for learning-dependent changes in functional brain activations during training. GLM analysis of the fMRI data across training runs showed significant changes in occipito-temporal BOLD across tasks (*Figure 1—figure supplement 2*), suggesting that BOLD changes at this early stage of learning (i.e. single training session that resulted in maximum 74% mean performance) do not differ between tasks (main effect of Task: $F_{(1,34)}= 0.20$, p=0.66; Task x Block interaction: $F_{(1.9,64.6)}$, p=0.71). This is consistent with previous fMRI studies showing learning-dependent changes within a single training session (*Mukai et al., 2007*). It is possible that the two tasks may show discriminable BOLD activations after more extensive training resulting in saturated performance, as shown by our previous studies using similar learning paradigms with multiple training sessions (*Kourtzi et al., 2005*; *Li et al., 2012*; *Mayhew et al., 2012*). Second, we conducted whole-brain voxel-wise covariance analyses using either learning rate or Δd' as covariates. For these analyses, we pooled the data across tasks, as changes in both behavioral performance and BOLD with training were similar between tasks. Our results showed significant correlations between BOLD change (late vs. early training runs) in the posterior occipito-temporal cortex and behavioral improvement (learning rate, Δd') across tasks (*Figure 1c*, *Figure 1—figure supplement 3*, *Figure 1—source data 1*). These results provide evidence for learning-dependent changes in occipito-temporal cortex that relate to behavioral improvement, consistent with our previous studies and the role of this region in visual learning and global shape processing (*Kourtzi et al., 2005*; *Kuai et al., 2013*; *Zhang et al., 2010*). Therefore, we next focused on the posterior occipito-temporal cortex and tested whether learning-dependent BOLD changes relate to changes in GABA concentration in this region.

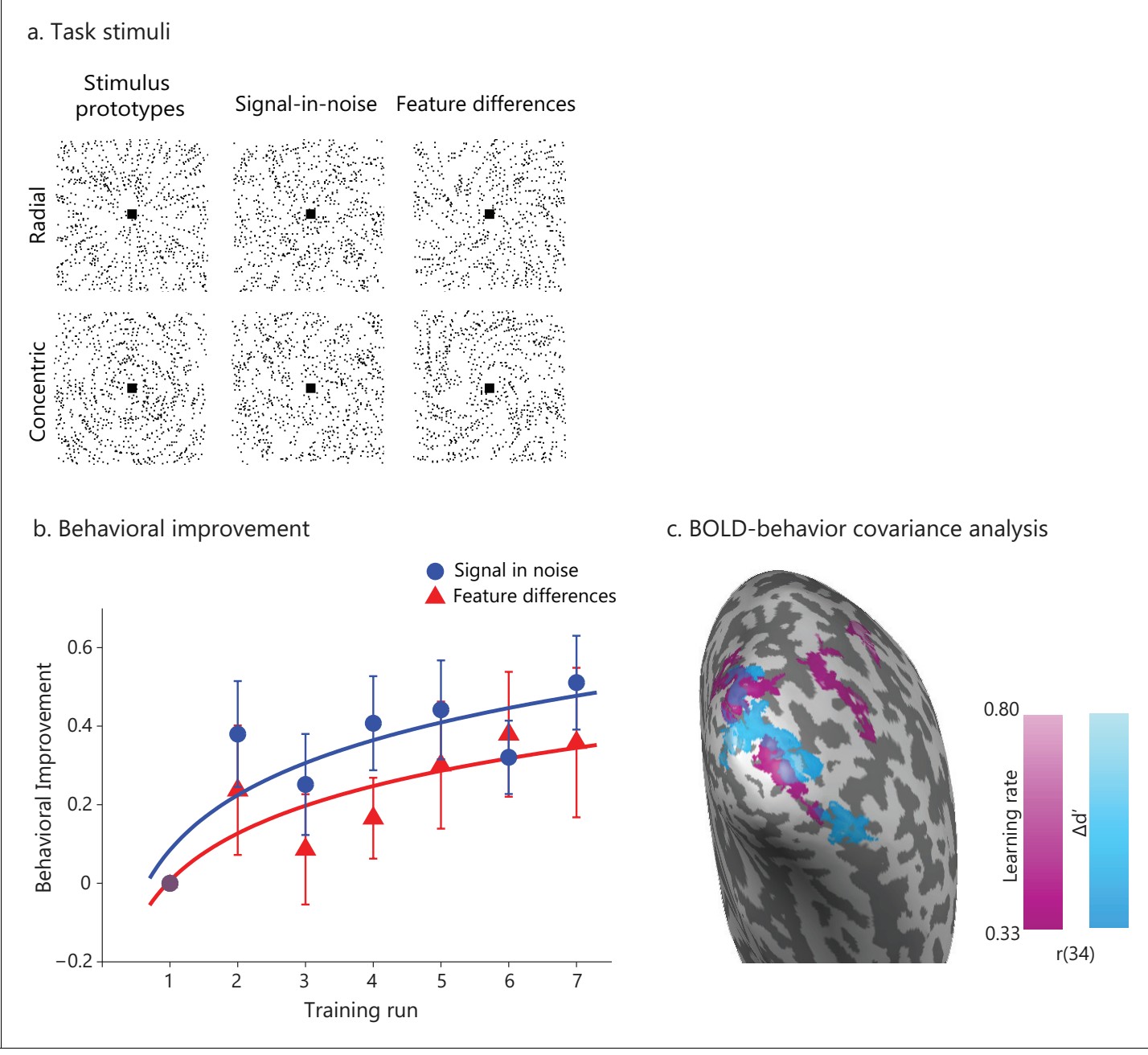

**Figure 1.** Learning-dependent changes in behavior and brain activation. (**a**) Stimuli: Example stimuli comprising radial and concentric Glass patterns. Stimuli are shown for the Signal in noise task (25% signal, spiral angle 0° for radial and 90° for concentric) and the Feature-differences task version (100% signal, spiral angle 38° for radial and 52° for concentric). Prototype stimuli (100% signal, spiral angle 0° for radial and 90° for concentric) are shown for illustration purposes only. (**b**) Behavioral improvement during training: mean d' per training run normalized to d' in the first run. Data were fitted with a logarithmic function; error bars indicate standard error of the mean across participants. The trend of higher performance in the SN than the FD task was not statistically significant. No significant improvement was observed for a no-training control group who did not receive training in between test sessions (*Figure 1—figure supplement 1*). (**c**) Whole-brain covariance analyses (cluster threshold corrected, p<0.05) with either learning rate (magenta) or Δd' (blue) on fMRI data (first two runs vs. last two runs) that were pooled across the two tasks showed positive significant clusters in the posterior occipito-temporal cortex. Activations are shown on the cortical surface of the right hemisphere (sulci are shown in dark grey, gyri in light grey). The color bar indicates Pearson's r correlation values. *Figure 1—figure supplement 3* illustrates the relationship between BOLD change extracted from this region and measures of behavioral improvement (learning rate, Δd') per task. Significant activations were observed in bilateral occipito-temporal cortex and fronto-parietal regions (*Figure 1—source data 1*). Further, GLM analysis of the fMRI data across training runs showed significant changes in occipito-temporal BOLD for both tasks (*Figure 1—figure supplement 2*). For all figures, data are included for the same training duration across participants (i.e. seven runs), as several participants (n = 9) were missing data from the eighth run. Including data from participants that were trained for

*Figure 1 continued on next page*

*Figure 1 continued*

an additional eighth run (*Figure 1—figure supplement 4a*) showed similar results as the analysis including seven training runs from all participants; that is, the whole brain covariance analysis of BOLD change with behavioural improvement showed similar activation maps (*Figure 1—figure supplement 4b,c*).

DOI: https://doi.org/10.7554/eLife.35854.003

The following source data and figure supplements are available for figure 1:

**Source data 1.** Tables for whole brain GLM covariance analysis of BOLD with behavioral improvement.

DOI: https://doi.org/10.7554/eLife.35854.008

**Figure supplement 1.** No training control group.

DOI: https://doi.org/10.7554/eLife.35854.004

**Figure supplement 2.** BOLD changes during training.

DOI: https://doi.org/10.7554/eLife.35854.005

**Figure supplement 3.** Relating BOLD change to behavioral improvement.

DOI: https://doi.org/10.7554/eLife.35854.006

**Figure supplement 4.** Behavior and brain imaging analyses including data from the eighth training run.

DOI: https://doi.org/10.7554/eLife.35854.007

## Relating GABA to behavioral improvement

Previous MRS studies have shown that GABA concentrations in the visual cortex relate to performance in perceptual tasks (*Edden et al., 2009*) and homeostatic plasticity (*Lunghi et al., 2015*). Here, we test whether GABAergic inhibition relates to behavioral improvement and learning-dependent functional changes in the visual cortex, by comparing MRS-measurements of GABA in the posterior occipito-temporal cortex before vs. after training.

First, we tested whether behavioral improvement –as measured by learning rate and Δd'– relates to changes in visual cortex GABA with training. We recorded GABA concentrations before and after training within a voxel centered on the posterior-occipito-temporal cortex (*Figure 2—figure supplement 1*), consistent with the fMRI analysis showing learning-dependent BOLD changes with training in this region. Correlating learning rate and Δd' with GABA changes showed dissociable effects for the two tasks (*Figure 2*; *Figure 2—figure supplement 4*). In particular, for the Signal-in-noise task we observed a negative correlation of GABA change with learning rate (r = −0.43, CI=[−0.74,−0.07]), but no significant correlation with Δd' (r = −0.14, CI=[−0.49, 0.29]). In contrast, for the Feature-differences, task we observed a positive correlation of GABA change with Δd' (r = 0.54, CI=[0.05, 0.85]), but no significant correlation with learning rate (r = 0.13, CI=[−0.38, 0.62]). Further, the significant correlations of GABA change with behavioral improvement (learning rate for SN; Δd' for FD) were significantly different between tasks (Fisher's z = 2.91, p=0.004). These dissociable effects could not be simply explained by differences in overall performance between tasks, as the two tasks resulted in similar behavioral improvement.

To ensure that our results were specific to GABA changes in the posterior occipito-temporal cortex due to training, we performed the following controls. First, correlation of GABA change and behavioral improvement remained significant when we corrected for a) tissue (grey matter, white matter, cerebrospinal fluid) composition (SN, correlation with learning rate: r = −0.41, CI=[−0.70,−0.07]; FD, correlation with Δd': r = 0.56, CI=[0.03, 0.83]) and b) differences in data quality (as measured by Cramer-Rao Lower Bounds – see Materials and methods) between the two GABA measurements (SN, correlation with learning rate: r = −0.44, CI=[−0.71,−0.12]; FD, correlation with Δd': r = 0.46, CI=[0.03, 0.76]). Second, correlating percentage GABA change (GABA change/pre training GABA) with behavioral improvement to control for pre-training GABA showed significant correlations for both tasks (SN, correlation with learning rate: r = −0.45, CI=[−0.78,−0.002]; FD, correlation with Δd': r = 0.58, CI=[0.18, 0.81]). These correlations were significantly different between tasks (Fisher's z = 2.87, p=0.004) and remained so when we referenced GABA to NAA rather than creatine concentration (Fisher's z = 2.73, p=0.01). Third, changes in Glutamate, the other major cortical neurotransmitter, did not correlate significantly with behavioral improvement (SN, correlation of Glutamate change with learning rate: r = 0.33, CI=[−0.22, 0.67]; FD, correlation of Glutamate change with Δd': r = −0.30, CI=[−0.58, 0.06]). These correlations of Glutamate change with measures of behavioral improvement were significantly different from correlations of GABA change with behavioral improvement (SN, correlations with learning rate: Steiger's z = 2.99, p=0.003; FD, correlations with

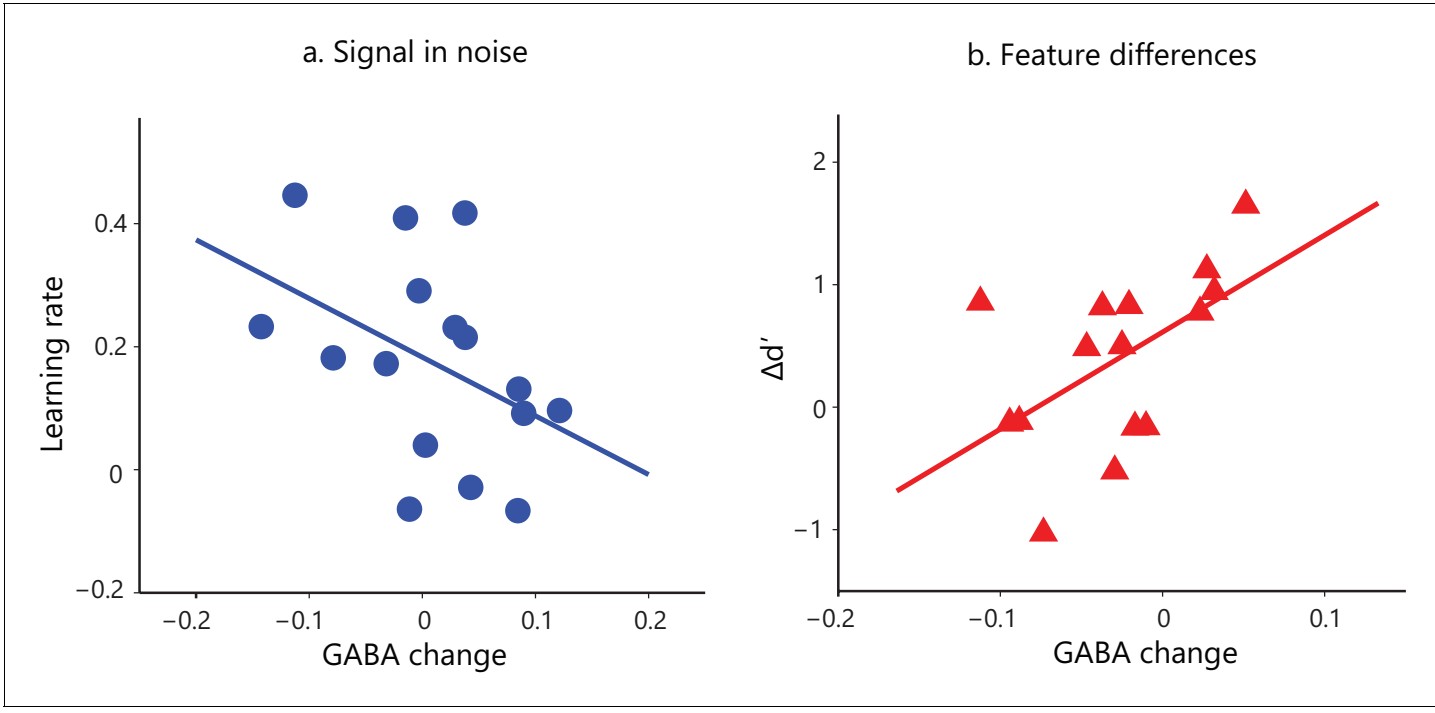

**Figure 2.** Task-dependent correlations of GABA change and behavioral improvement. We measured MRS GABA in the posterior occipito-temporal cortex (*Figure 2—figure supplement 1*). We pre-processed and fit the data with ProFit (*Schulte and Boesiger, 2006*). We excluded three participants (FD task) due to fat contamination in the spectra (*Figure 2—figure supplement 2*). We did not observe significant differences in mean GABA concentration in occipito-temporal cortex before vs. after training (*Figure 2—figure supplement 3*). Here, we show skipped Pearson's correlations indicating (a) a significant negative correlation of GABA change in occipito-temporal cortex with learning rate for the Signal-in-noise task (r = −0.43, CI=[−0.74,–0.07]) and (b) a significant positive correlation with Δd' for the Feature-differences task (r = 0.54, CI=[0.05,0.85]). Correlations of GABA change with Δd' for the Signal-in-noise task or learning rate for the Feature-differences task were not significant (*Figure 2—figure supplement 4a*). Negative learning rate or negative Δd' represents decreased sensitivity during training. As participants were trained only for a single training session and without trial-by-trial feedback, these measures may be noisier and result in negative values. Correlations of GABA change with behavioral improvement remained significant when we removed data from participants with negative learning rate or Δd'. Further, including data from participants that were trained for an additional eighth run showed similar results: the correlations of GABA change with behavioral improvement (learning rate computed with eight training runs; Δd' (last vs. first training run) remained significantly different (Fisher's z = 2.4, p=0.02) between tasks (*Figure 2— figure supplement 4b*).

DOI: https://doi.org/10.7554/eLife.35854.009

The following source data and figure supplements are available for figure 2:

**Source data 1.** GABA change, learning rate and Δd' per participant.

DOI: https://doi.org/10.7554/eLife.35854.014

**Figure supplement 1.** MRS voxel placement.

DOI: https://doi.org/10.7554/eLife.35854.010

**Figure supplement 2.** Fat contamination.

DOI: https://doi.org/10.7554/eLife.35854.011

**Figure supplement 3.** GABA concentration before vs. after training.

DOI: https://doi.org/10.7554/eLife.35854.012

**Figure supplement 4.** Correlations of GABA change and behavioral improvement.

DOI: https://doi.org/10.7554/eLife.35854.013

Δd': Steiger's z = 3.34, p=0.001). Further, correlations of GABA change and behavioral improvement remained significant after accounting for Glutamate change (SN, correlation of GABA change with learning rate: r = −0.41, CI=[−0.69,–0.08]; FD, correlation of GABA change with Δd': r = 0.54, CI= [0.04, 0.85]), suggesting that our results were specific to GABA and do not generalize to glutamate. Finally, to test whether our findings were specific to occipito-temporal cortex GABA, we measured learning related GABA changes in both the posterior occipito-temporal cortex and the posterior parietal cortex (IPS) in an independent group of participants (SN: n = 17; FD: n = 21). We found

significant and opposite correlations of occipito-temporal GABA change with behavioral improvement for the two tasks (SN: r = −0.43, CI=[−0.75,–0.02]; FD: r = 0.55, CI=[0.10, 0.78]). We did not find significant correlations between posterior parietal GABA change and behavioral improvement for either task (SN: r = −0.23, CI=[−0.61, 0.19]; FD: r = 0.05, CI=[−0.37, 0.43]), suggesting that our findings are specific to local changes in occipito-temporal GABA rather than reflecting changes in global cortical excitability.

Our analyses so far showed significant correlations of changes in GABA and behavior due to training. Yet, we did not observe significant differences in mean GABA concentration in occipito-temporal cortex before vs. after training (main effect of MRS block: F(1,34)= 0.06, p=0.81; Task x MRS block interaction: F(1,34)= 0.21, p=0.65) (*Figure 2—figure supplement 3*). Previous studies have reported mean changes in GABA concentration in the motor cortex (*Floyer-Lea et al., 2006*; *Sampaio-Baptista et al., 2015*) due to training and visual cortex due to changes in homeostatic plasticity (*Lunghi et al., 2015*). The main difference between our study and these previous reports is that participant performance increased but did not saturate during the single training session employed in our study (i.e. participant reached mean performance 74%), in contrast to previous studies that showed saturated performance after training. Thus, it is likely that mean changes in GABA concentration are more pronounced when participant performance has plateaued after training. Further, it is likely that 7T imaging (rather than 3T imaging used in our study) affords increased signal-to-noise ratio and time resolution that may benefit measurements of change in GABA concentration (*Barron et al., 2016*; *Lunghi et al., 2015*).

## Relating GABA to learning-dependent BOLD change

Next, we tested whether learning-dependent changes in visual GABA (before vs. after training) relate to changes in BOLD within the posterior occipito-temporal cortex. We conducted a GLM covariance analysis to test whether BOLD changes (late vs. early training runs) relate to GABA changes in this region. This analysis showed opposite correlations between GABA and BOLD change for the two tasks: negative correlation for the Signal-in-Noise, while positive correlation for the Feature-differences task (*Figure 3a*). We corroborated this result by extracting BOLD signal from the voxel clusters in the posterior occipito-temporal cortex that resulted from the covariance analysis of fMRI with behavioral improvement (*Figure 1c*). Correlations of change in GABA and BOLD – extracted from this independently defined region of interest (*Figure 1c*)- were opposite and significantly different between the two tasks (SN: r = −0.58 CI=[−0.82,–0.22], FD: r = 0.70 CI=[0.37, 0.90], Fisher's z = 4.19, p<0.0001) (*Figure 3b*).

Our findings suggest that task-dependent suppression mechanisms relate to functional changes in visual cortex and behavioral improvement. To further test this hypothesis, we performed moderation analyses (*Hayes, 2012*) (*Figure 3—figure supplement 2*) that allowed us to test whether the influence that an independent variable (i.e. BOLD) has on the outcome (i.e. behavior) is moderated by one or more moderator variables (i.e. GABA, task). Our results showed that this model is significant (F(7,28)=3.77, p=0.01) and the relationship between BOLD change and behavioral improvement depends multiplicatively on GABA change and task, as indicated by a significant three-way interaction between task, GABA change, and BOLD change (F(1,28)= 7.17, p=0.01; R-square change = 0.13). These results suggest that task-dependent GABAergic inhibition moderates the relationship between functional brain plasticity and behavioral improvement in the visual cortex.

## Control analyses

To ensure that the dissociable correlations we observed between tasks for behavior, GABA and BOLD were not due to differences between the two groups of participants that were each trained on a different task (SN vs FD group), we compared behavioral and imaging data between groups before training. First, our analyses did not show any significant differences in GABA concentration before training (t(34)=0.11, p=0.91) nor in behavioral performance early in training (i.e. first training run) (t(34)=0.23, p=0.82) between the two groups. Second, we compared signal-to-noise ratio (SNR) between tasks for the first MRS measurement (i.e. pre-training) and the first two fMRI runs (i.e. early in the training, as there were no fMRI measurements before training). We did not find any significant differences in MRS SNR (t(34)=0.77, p=0.45), nor fMRI temporal SNR (tSNR) between the two tasks

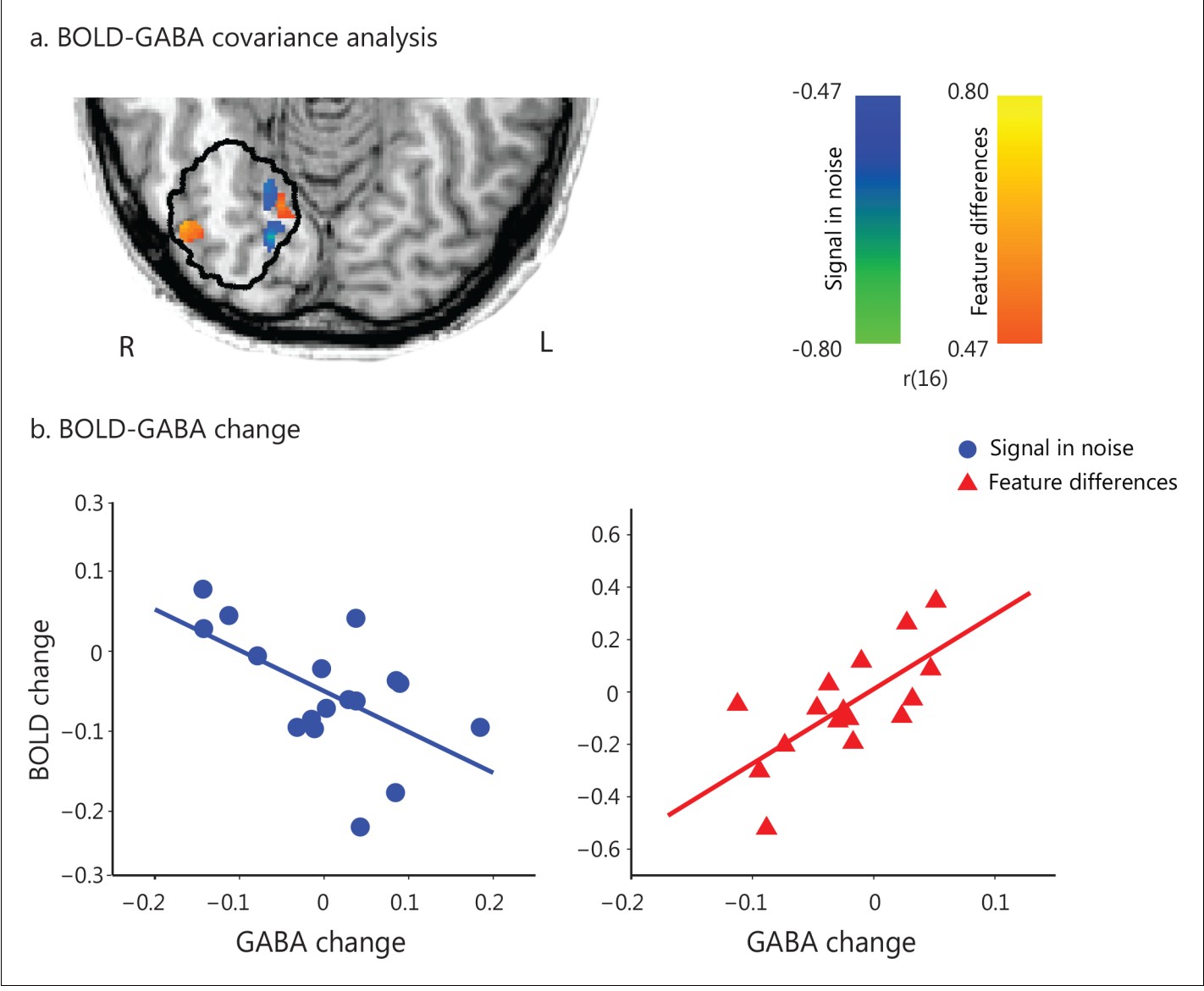

**Figure 3.** Task-dependent correlations of GABA change with BOLD change. (**a**) GLM covariance analysis of GABA change with BOLD change within a masked region defined by the MRS voxel probability map (i.e. gray matter voxels within each participant's MRS voxels with minimum 50% probability, as outlined in black). We used fMRI data (i.e. first two vs. last two fMRI runs) that were collected closer to the time when GABA was collected (before vs. after training). Activations are shown in radiological co-ordinates. GABA change correlated negatively with BOLD change for the Signal-in-Noise task (green to blue color bar), while positively for the Feature-differences task (orange to yellow color bar). The color bars indicate Pearson's r. (**b**) Correlation of change in GABA and BOLD extracted from an independently defined region of interest; that is BOLD was extracted from the voxel clusters in posterior occipito-temporal cortex that resulted from the covariance analysis of fMRI with behavioral improvement (*Figure 1c*). This analysis showed opposite and significantly different correlations (SN: r = −0.58 CI=[−0.82,–0.22], FD: r = 0.70 CI=[0.37, 0.90], Fisher's z = 4.19, p<0.0001) and corroborated the results shown in *Figure 3a*. Including data from participants that were trained for an additional eighth run showed similar results, as the analyses including seven training runs from all participants (*Figure 3—figure supplement 1*). In particular, a) the whole brain covariance analysis of BOLD change with GABA change showed similar activation maps (*Figure 3—figure supplement 1a*), b) the correlations between GABA change and BOLD change (extracted from the voxels activated in the independent covariance analysis with behavioral improvement, *Figure 1—figure supplement 4b*) remained significantly different between tasks (Fisher's z = 3.26, p=0.001, *Figure 3—figure supplement 1b*). Finally, moderation analysis showed a significant interaction between GABA change, task and BOLD change (*Figure 3—figure supplement 2*).

DOI: https://doi.org/10.7554/eLife.35854.015

The following source data and figure supplements are available for figure 3:

**Source data 1.** GABA change and BOLD change (from the voxel clusters in posterior occipito-temporal cortex, *Figure 1c*) per participant.

DOI: https://doi.org/10.7554/eLife.35854.018

**Figure supplement 1.** Correlations of GABA change and BOLD change for Signal-in-Noise vs Feature differences task including eighth training run.

*Figure 3 continued on next page*

*Figure 3 continued*

DOI: https://doi.org/10.7554/eLife.35854.016

**Figure supplement 2.** Task-dependent GABAergic plasticity moderates the relationship of functional brain plasticity and behavioral improvement.
DOI: https://doi.org/10.7554/eLife.35854.017

(t(34)=0.73, p=0.47). These results suggest that the dissociable results we observed between tasks could not be simply due to differences in the two groups.

Further, to ensure that the learning-dependent changes we observed were not confounded by changes in the scanner environment during training, we conducted the following control analyses. First, we calculated the variation of the scanner center frequency across training runs for each participant. We found that the mean scanner center frequency variation across participants was very small (mean and standard deviation across participants: 0.0000125 ± 0.0000019 MHz), and there was no significant interaction between Training (first two vs. last two fMRI runs) and Task (F(1,34)=0.68, p=0.42). Second, a similar analysis on tSNR across fMRI runs did not show a significant interaction between Training (first two vs. last two fMRI runs) and Task (F(1,34)= 1.62, p=0.21). Further, to control for measurement differences in the MRS before vs. after training we conducted the following analyses. First, to assess measurement quality we calculated spectral SNR for each MRS measurement. This analysis showed no significant interaction between MRS block and task (F(1,34) = 2.37, p=0.13) nor a main effect of block (F(1,34)= 1.60, p=0.22). Second, to assess spectral resolution before vs. after training, we calculated peak linewidth for each MRS measurement. This analysis showed no significant interaction between MRS block and task (F(1,34)=0.90, p=0.35) nor a significant main effect of MRS block (F(1,34)= 2.97, p=0.09). These results suggest that the MRS data quality was similar before and after training for both tasks. Taken together these analyses suggest that the dissociable correlations between BOLD and GABA we observed between tasks could not be due to differences in the quality of the BOLD or GABA measurements during training.

Finally, to ensure that our results were specific to learning-dependent changes, we excluded data from participants who did not show positive improvement during the single training session employed in our study, as indicated by learning rate (n = 3) or Δd′ (n = 8). Despite the smaller data sample, the following results remained significant: a) correlations of GABA change with behavioral improvement (SN: r = −0.52, CI=[−0.80,–0.09]; FD: r = 0.72, CI=[0.29, 0.94]), b) correlations of BOLD change (early vs. late training runs) with behavioral improvement (learning rate (r = 0.57, CI=[0.24, 0.80]) and with Δd′ (r = 0.67, CI=[0.33, 0.86]). Further, the correlations between GABA change and BOLD change (extracted from the voxel clusters revealed by the independent covariance analysis with behavioral improvement) remained significantly different between tasks (z = 2.84, p=0.01).

## Brain stimulation modulates behavioral improvement during training

To extend beyond correlative evidence, we sought to perturb cortical excitability using transcranial direct current stimulation (tDCs) that has been previously shown to alter overall responsivity of the visual cortex (i.e. modulate visual evoked potentials) (*Antal et al., 2004a*). Our findings on the relationship of GABA change and behavioral improvement lead to opposite predictions for the effect of tDCs on the two learning tasks. In particular, anodal tDCs is known to be excitatory (*Nitsche and Paulus, 2000*) and has been shown to result in local GABA reduction in visual (*Barron et al., 2016*) and motor cortex (*Stagg et al., 2009*). Further, anodal tDCs has been shown to facilitate learning in motor (*O'Shea et al., 2017*; *Stagg et al., 2011c*) and perceptual tasks (*Fertonani et al., 2011*; *Pirulli et al., 2013*; *Sczesny-Kaiser et al., 2016*). Our results for the Signal-in-Noise task showed that GABA change correlated negatively with behavioral improvement, suggesting that decreased GABA relates to higher behavioral improvement. Therefore, we hypothesized that excitatory anodal tDCs would enhance performance during training on this task. In contrast, cathodal stimulation is thought to be inhibitory; that is, it has been shown to reduce cortical excitability (*Nitsche and Paulus, 2000*) by decreasing glutamatergic transmission (*Stagg et al., 2009*). Further, cathodal tDCs on the occipital cortex has been shown to facilitate performance in perceptual judgments by suppressing incorrect sensory input (*Antal et al., 2004b*). Our results for the Feature differences task showed that GABA change correlated positively with behavioral improvement, suggesting that increased GABA relates to higher behavioral improvement. Therefore, we hypothesized that the inhibitory

cathodal –rather than the excitatory anodal– stimulation would enhance performance during training on this task.

To test these hypotheses, participants in different groups received anodal, cathodal or sham stimulation in the posterior occipito-temporal cortex during training on the Signal-in-Noise or the Feature-differences task. Our results (*Figure 4*) showed significant improvement in behavioral performance, as measured by d', for anodal compared to sham stimulation for the Signal-in-Noise (Training block x Stimulation: $F_{(2,52.5)}$ = 3.99, p=0.02) but not the Feature-differences task (Training block x Stimulation: $F_{(2.2,57.9)}$ = 0.45, p=0.66). In contrast, we observed improved performance during cathodal compared to sham tDCs for the Feature-differences task (main effect of Stimulation: $F_{(1,26)}$= 6.13, p=0.02) but not the Signal-in-Noise task (main effect of Stimulation: $F_{(1,26)}$= 0.001, p=0.98). To compare behavioral improvement between tasks, we normalized performance during tDCs (anodal or cathodal) to performance during sham stimulation (*Figure 4a*). A repeated-measures

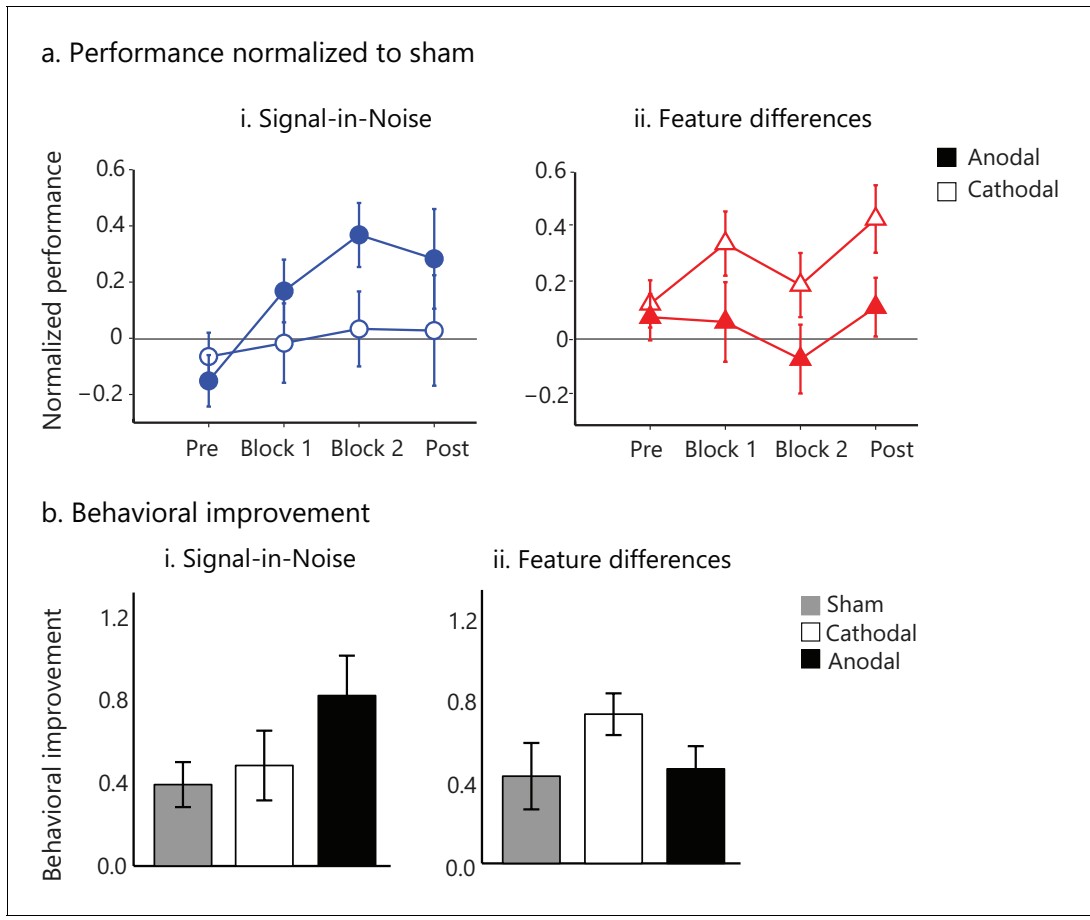

**Figure 4.** tDCs intervention facilitates visual learning. (a) Task performance (d') for the active stimulation groups (anodal, cathodal tDCS on posterior occipito-temporal cortex (*Figure 4—figure supplement 1*)) normalized to the sham group for the pre- and post- training blocks (no feedback, no stimulation) and the two training blocks (Block 1, Block 2; 500 trials per block). Performance (d') was significantly enhanced for anodal (but not cathodal) stimulation in the Signal-in-Noise task, while for cathodal (but not anodal) stimulation in the Feature differences task. (b). Behavioral improvement (d' post- minus pre-training) was enhanced for the anodal stimulation group in the Signal-in-Noise task and the cathodal stimulation group for the Feature differences task. Error bars indicate standard error of the mean across participants.

DOI: https://doi.org/10.7554/eLife.35854.019

The following source data and figure supplement are available for figure 4:

**Source data 1.** Performance (d') normalized to the sham group for each participant of the anodal and cathodal groups for the two tasks.
DOI: https://doi.org/10.7554/eLife.35854.021
**Figure supplement 1.** tDCs electric field simulation.
DOI: https://doi.org/10.7554/eLife.35854.020

ANOVA showed a significant Task, Stimulation x Training block interaction (F(2.5, 130)=3.19, p=0.03).

We next conducted the following control analyses to ensure that our results relate to learning-dependent changes in behavioral performance rather than differences in task difficulty across stimulation groups. Comparing performance across participants before training (i.e. pre-training block with no feedback or stimulation) showed no significant effect of Task (F(1,78)=0.05, p=0.82), nor a significant interaction between Task and Stimulation group (F(2,78)=0.92, p=0.40), suggesting that the tDCs-induced learning effects were not due to differences in difficulty across tasks (*Figure 4*). This was further supported by a significant main effect of Session (F(1,78)=86.99, p<0.0001) across stimulation groups suggesting that all participants (including the sham stimulation groups) were able to learn the task (*Figure 4*). Further, the double dissociation we observed between task and stimulation site makes it unlikely that stimulation could produce a non-specific effect on general behavioral performance (e.g., through distraction caused by skin irritation). In contrast, comparing performance on consecutive days in a no-training control group (participants were tested twice but without training on the task) showed no significant main effect of Session (F(1,17)= 0.78, p=0.39) nor a significant interaction between Task and Session (F(1,17)=0.30, p=0.59), suggesting that the learning effects we observed were training-specific. Finally, to test whether behavioral improvement was maintained after training, we compared performance in the last training block (feedback, stimulation) vs. a post-training test that was conducted on the day following training (no feedback, no stimulation). A repeated-measures ANOVA showed no significant interaction between Session x Stimulation x Task (F(2,78)=1.09, p=0.34), suggesting that improved performance was maintained across all groups when participants were tested without tDCs stimulation.

Together, these results demonstrate dissociable effects of tDCs stimulation on behavioral improvement between tasks, suggesting that GABAergic inhibition alters learning and experience-dependent plasticity in the posterior occipito-temporal cortex. In particular, we demonstrate that excitatory stimulation enhances performance during training to detect targets from noise, while inhibitory stimulation enhances fine feature discriminability. These results are consistent with the opposite correlations of change in occipito-temporal GABA and behavioral improvement that we observed across tasks. Taken together, our findings suggest that GABAergic processing in visual cortex optimizes noise filtering for target detection, while retuning of feature templates for fine discrimination.

## Discussion

Here, we demonstrate that GABAergic inhibition relates to dissociable learning mechanisms that mediate improved perceptual decisions under uncertainty: when learning to detect targets embedded in clutter or discriminate between highly similar features. Interrogating fMRI signals alone does not allow us to discern between the brain mechanisms that underlie these skills, as BOLD reflects the aggregate activity of excitatory and inhibitory signals at the scale of large neural populations (*Heeger and Ress, 2002*; *Logothetis, 2008*). Our results showed similar learning dependent changes in behavioral performance and BOLD in these tasks at early stages of learning (i.e. training for a single session). This is consistent with previous fMRI studies of perceptual learning that have shown learning-dependent changes in the overall fMRI responses in visual cortex (e.g. [*Kourtzi et al., 2005*; *Mukai et al., 2007*; *Sigman et al., 2005*]), or enhanced discriminability of fMRI patterns with training (*Byers and Serences, 2014*; *Jehee et al., 2012*; *Kuai et al., 2013*; *Zhang et al., 2010*).

However, combining MRS measurements of GABA with fMRI uncovers distinct suppression mechanisms that moderate the relationship between behavioral improvement and experience-dependent plasticity in visual cortex. Previous studies have investigated the relationship of baseline GABA measurements with performance in the context of visual (*Edden et al., 2009*) and sensory-motor tasks (*Heba et al., 2016*; *Kolasinski et al., 2017*; *Stagg et al., 2011a*) as well as reward-based learning (*Scholl et al., 2017*). Here, we test whether learning-dependent changes in GABA (i.e. GABA changes before vs. after training) relate to changes in performance (i.e. behavioral improvement) and functional activation. Previous studies have reported changes in GABA within the range and time scales observed in our study (10–15% change observed within 20–30 min) due to stimulation (*Barron et al., 2016*; *O'Shea et al., 2017*; *Stagg et al., 2009*) or training (*Floyer-Lea et al., 2006*)

suggesting a role for GABAergic inhibition across stages of learning (*Sampaio-Baptista et al., 2015*; *Shibata et al., 2017*). Although, the precise mechanisms that underlie changes in GABA as measured by MRS are still under investigation (*Stagg, 2014*), recent animal (*Mason et al., 2001*) and human (*Stagg et al., 2011b*) studies suggest that MRS-measured GABA reflects primarily extra-synaptic GABA concentrations.

Our findings provide evidence that changes in GABAergic inhibition in the visual cortex relate to learning-dependent changes in behavior and functional brain plasticity. In particular, for learning to see in clutter, decreased occipito-temporal GABA relates to increased BOLD and improved performance, as indicated by faster learning rate. These findings suggest enhanced noise filtering through gain control that is mediated by learning-dependent changes in visual cortex suppression. This mechanism is consistent with previous animal work linking GABAergic inhibition to neural gain (*Mitchell and Silver, 2003*) and interventional studies showing that blocking GABAergic inhibition increases neural gain (*Hamann et al., 2002*). It is possible that learning to detect targets in clutter is implemented by decreased local suppression that facilitates recurrent processing for noise filtering and target detection (*Gilbert and Li, 2012*; *Poort et al., 2016*). In contrast, we demonstrate that for learning to discriminate fine feature differences, increased occipito-temporal GABA relates to increased BOLD and improved performance, as indicated by enhanced sensitivity in visual discrimination after training. This finding suggests that learning to discriminate between highly similar targets by template retuning involves neural tuning to fine feature differences (*Raiguel et al., 2006*; *Schoups et al., 2001*; *Yang and Maunsell, 2004*) that is mediated by increased visual cortex suppression. This mechanism is consistent with studies linking GABAergic inhibition to cortical tuning (*Rokem et al., 2011*; *Wehr and Zador, 2003*) and pharmacological interventions showing that GABA agonists enhance orientation selectivity in visual cortex (*Leventhal et al., 2003*; *Li et al., 2008*), while blocking GABAergic inhibition results in broader neural tuning (*Leventhal et al., 2003*; *Sillito, 1979*).

Our results showed an intriguing dissociation between tasks; that is, learning rate (but not $\Delta d'$) correlated significantly with GABA change for the Signal-in-Noise task, while $\Delta d'$ (but not learning rate) correlated significantly with GABA change for the Feature differences task. Recent studies characterizing the role of different populations of interneurons in visual learning may shed light into this task-dependent GABAergic plasticity. In particular somatostatin-positive (SOM) interneurons have been implicated in spatial summation (*Adesnik et al., 2012*) and have been shown to gate plasticity during training by providing contextual information (*van Versendaal and Levelt, 2016*). In contrast, parvalbumin-positive (PV) interneurons have been implicated in selective inhibition (*Rokem et al., 2011*) that sharpens feature representations after training (*Khan et al., 2018*). It is therefore possible, that the dissociable correlations we observed between tasks for GABA change and behavioral improvement may reflect differential involvement of SOM vs. PV interneurons in the two tasks. Specifically, SOM interneurons involved in spatial integration may support learning to detect targets from clutter (SN task) through noise filtering. In contrast, PV interneurons involved in selective inhibition may support learning fine differences (FD task) through re-tuning of feature templates. Further, SOM vs. PV interneurons are shown to be involved at different stages during the time course of learning. In particular, SOM cells have been shown to gate learning-dependent plasticity during training (*Chen et al., 2015*), while PV cells form stimulus-specific ensembles with pyramidal cells after training on a visual discrimination task (*Khan et al., 2018*). Thus, it is possible that different behavioral measures capture the function of SOM vs. PV interneurons, consistent with the dissociation we observed between tasks for the correlations of GABA change and behavioral improvement. In particular, learning rate (i.e. the rate with which perceptual sensitivity changes during training) may capture best the function of SOM interneurons that act during learning to support noise filtering throughout the course of training. In contrast, $\Delta d'$ (i.e. change in perceptual sensitivity after training) may capture best the function of PV interneurons that are shown to support tuning of stimulus-specific representations after training.

Finally, to directly test whether visual cortex suppression mediates behavioral improvement due to training, we employed tDCs to perturb cortical excitability. Our results demonstrate a double dissociation in learning-dependent mechanisms of visual plasticity. Excitatory anodal (rather than cathodal) stimulation enhanced learning to detect targets in clutter, consistent with the negative correlation between GABA change and behavior . In contrast, inhibitory cathodal (rather than anodal) stimulation enhanced learning to discriminate fine features, consistent with the positive

correlation between GABA change and behavior . These findings demonstrate a direct link between GABAergic processing in visual cortex and enhanced visual learning.

In sum, our findings provide novel evidence for the role of GABAergic processing in learning, shedding light on the neural implementation of theoretical models of perceptual learning. Future animal studies may probe the micro-circuits that give rise to learning by noise filtering vs. feature template retuning. Recent work has begun to classify cortical interneurons into distinct classes based on morphology, connectivity, and physiology (*Kepecs and Fishell, 2014*) and link them to distinct cortical computations (see for example (*El-Boustani and Sur, 2014*; *Kerlin et al., 2010*; *Wilson et al., 2012*). These distinct interneuron types may differentially contribute to learning by noise filtering vs. feature template retuning by changing the gain vs. feature selectivity of pyramidal cells. Thus, our findings propose testable hypotheses linking theoretical models of perceptual learning to the micro-circuits that mediate adaptive behavior and underlie the macroscopic learning-dependent plasticity, as measured by human brain imaging.

# Materials and methods

## Participants

A hundred and thirty participants took part in this study. Forty six participants (21 female; mean age 25.04 ± 3.69 years) participated in the brain imaging experiment and eighty four participants (45 female; mean age 23.8 ± 3.41 years) participated in the brain stimulation experiment. All participants were right-handed, had normal or corrected-to-normal vision and gave written informed consent. The study was approved by the University of Cambridge ethics committee.

For the brain imaging experiment, we trained two separate groups of participants. Each group was trained only on one of the two tasks (SN, FD) to avoid transfer effects across tasks that have been previously reported when the same individuals were trained sequentially on both tasks (*Chang et al., 2013*; *Dosher and Lu, 2007*). Data from three participants were excluded from the study due to technical failure during data acquisition resulting in twenty one participants in the Signal-in-Noise experiment and twenty two participants in the Feature-differences experiment. Sample size was determined based on power calculations following previous studies on motor learning showing an effect size of r = 0.65 or r = 0.60 at 90% power for correlations of GABA with behavior or BOLD, respectively (*Stagg et al., 2011a*).

For the brain stimulation experiment, we randomly assigned participants into six groups of fourteen participants per group who trained on either the Signal-in-Noise or the Feature-differences task during online anodal, cathodal or sham stimulation. Sample size was determined based on power calculations following previous studies showing a polarity-specific stimulation effect for effect size of Cohen's d = 1.15 at 90% power (*Sczesny-Kaiser et al., 2016*). Previous tDCs studies on visual learning reported significant learning enhancement after stimulation for the same sample size as in our study (*Fertonani et al., 2011*; *Pirulli et al., 2013*).

## Stimuli

We presented participants with Glass patterns (*Glass, 1969*) generated using previously described methods (*Zhang et al., 2010*). In particular, stimuli were defined by white dot pairs (dipoles) displayed within a square aperture on a black background. For the brain imaging experiment stimuli (size=7.9° x 7.9°) were presented in the center of the screen. For the brain stimulation experiment, stimuli (size=7.9° x 7.9°), were presented at the left hemifield (11.6 arc min from fixation) contralateral to the stimulation site to ensure maximal effect of stimulation on stimulus processing. The dot density was 3%, and the Glass shift (i.e., the distance between two dots in a dipole) was 16.2 arc min. The size of each dot was $2.3 \times 2.3$ arc $min^2$. For each dot dipole, the spiral angle was defined as the angle between the dot dipole orientation and the radius from the center of the dipole to the center of the stimulus aperture. Each stimulus comprised dot dipoles that were aligned according to the specified spiral angle (signal dipoles) for a given stimulus and noise dipoles for which the spiral angle was randomly selected. The proportion of signal dipoles defined the stimulus signal level.

We generated radial (0° spiral angle) and concentric (90° spiral angle) Glass patterns by placing dipoles orthogonally (radial stimuli) or tangentially (concentric stimuli) to the circumference of a circle centered on the fixation dot. Further, we generated intermediate patterns between these two Glass

pattern types by parametrically varying the spiral angle of the pattern from 0° (radial pattern) to 90° (concentric pattern). We randomized the presentation of clockwise (0° to 90° spiral angle) and counterclockwise patterns (0° to −90° spiral angle) across participants. A new pattern was generated for each stimulus presented in a trial, resulting in stimuli that were locally jittered in their position.

For the Signal-in-Noise task, radial and concentric stimuli (spiral angle: 0° +- 90°) were presented at 24 ± 1% signal level; that is, 76% of the dipoles were presented at random position and orientation. For the Feature-differences task, stimuli were presented at 100% signal and spiral angle of ±38° (radial) or ±52° (concentric) (*Figure 1a*). To control for stimulus-specific training effects, we presented each participant with a newly generated set of stimuli. To control for potential local adaptation due to stimulus repetition, we jittered (±1–3°) the spiral angle across stimuli. These procedures ensured that learning related to global shape rather than local stimulus features.

## Experimental design
### Brain imaging experiment
All participants underwent fMRI scanning during training on either the Signal-in-Noise or the Feature-differences task. In addition, we recorded MRS-GABA data from occipito-temporal cortex before and after training. The day after the fMRI scanning, we tested participants on the trained task for 216 trials without feedback.

The fMRI measurements comprised 7–8 experimental runs (data were missing from several participants (n = 9) for the eighth training run). Each run lasted 330 s. We used an event related design and ensured that the order of trials was matched for history (two trials back) such that each trial was equally likely to be preceded by any of the conditions. The order of the trials differed across runs and participants. Two stimulus conditions (radial, concentric) and one fixation condition, with 36 trials per condition, were presented in each run. Each run comprised 110 trials (108 trials across conditions and two initial trials for balancing the history of the second trial) and two 9 s fixation periods at the beginning and end of the run. Participants were presented with a Glass pattern stimulus per trial and asked to judge whether the presented stimulus was radial or concentric. Participants received feedback on their average performance (i.e. percent correct) every 10–15 trials as indicated by a vertical color bar.

For fixation trials, a fixation dot was displayed in the center of the screen for 3 s. For experimental trials, a 200 ms stimulus presentation was followed by a 1300 ms fixation. After this fixed delay, a response cue appeared as either a red '+" or 'x'. If the response cue was a red '+", participants indicated radial versus concentric by pressing the left versus the right key. If the response cue was a red 'x', the opposite keys were used (e.g., radial = right key). This allowed us to dissociate the motor response (button press) from the stimulus related fMRI activations. The response cue remained on the screen for 1000 ms, followed by a fixation dot 500 ms before the next trial onset. Participants were familiarized with the task before scanning.

### Brain stimulation experiment
Participants were presented with Glass patterns stimuli and asked to judge whether the presented stimulus in each trial was radial or concentric. They participated in three experimental sessions. In the first session we measured task performance in one block of 90 trials without feedback or tDCs stimulation. In the second session, all participants were trained with trial-by-trial feedback for 1000 trials during tDCs stimulation (with a short 30 s break every 200 trials). On each trial, a stimulus was presented for 300 ms and was followed by fixation while waiting for the participant's response (button press). Participants were given trial-by-trial feedback on their responses followed by a fixation dot for 500 ms before the onset of the next trial. In the third session, we measured task performance in one block of 90 trials without feedback or tDCs stimulation.

## Data acquisition
### Brain imaging data acquisition
Experiments were conducted at the Cognition and Brain Sciences Unit, Cambridge (3T Magnetom Trio, Siemens). We collected T1-weighted anatomical data (MP RAGE; TR/TE/TI = 2250/2.98/900 ms; FOV = 256×256 × 192 mm; isotropic 1 mm) and echo planar imaging data (gradient echo-pulse sequences) from 27 slices (whole-brain coverage; TR, 1500 ms; TE, 29 ms; flip-angle, 78°; resolution

2.5 × 2.5×4 mm). Further, we acquired two MRS measurements (one before and one after training) using a 32-channel head coil. We centered the voxel for both MRS measurements (25 × 25 × 25 mm$^3$) in the right occipito-temporal cortex (*Figure 2—figure supplement 1*), as this region has been previously shown to be involved in template representation (*Chang et al., 2014*) and judgements of object properties (*Ellison and Cowey, 2006*). We positioned the MRS voxel manually using anatomical landmarks (Superior Temporal Gyrus, Middle Occipital Gyrus) on the acquired T1 scan to ensure that the voxel placement matched between the pre- and post- training measurements and across participants. We used a 2D 1H J-PRESS sequence (*Prescot and Renshaw, 2013*; *Schulte and Boesiger, 2006*) (TR/TE = 2000/31–229 ms; ΔTE = 2 ms (100 TE steps); four signal averages per TE step with online averaging; 2D spectral width = 2000×500 Hz, and 2D matrix size = 1024×100). Measurements with this sequence at 3T have been previously shown to be reliable and reproducible (*Prescot and Renshaw, 2013*; *Schmitz et al., 2017*). We conducted B0 shimming within the MRS voxel combining an automated phase map with interactive manual shimming until the full-width at half-maximum measured for the real component of the unsuppressed water signal was below 20 Hz. We placed six saturation bands at least 1 cm away from the cubic MRS voxel faces to suppress outer volume (OVS), using hyperbolic secant adiabatic full passage RF pulses. OVS was interleaved with water suppression via a WET scheme (*Prescot and Renshaw, 2013*). Water unsuppressed 2D 1H MRS data were also collected and used for eddy current correction.

## tDCS data acquisition

We used a multi-channel transcranial electrical stimulator (neuroConn DC-STIMULATOR MC, Ilmenau, Germany) to deliver anodal, cathodal or sham stimulation. We used a pair of rubber electrodes (3 × 3 cm$^2$ stimulating electrode, 5 × 5 cm$^2$ reference electrode), placed in square sponges that had soaked in saline. In the anodal and cathodal conditions, 1mA current was ramped up over 10 s, was held at 1mA for 35 min and was subsequently ramped down over 10 s. In the sham condition, the current ramped up (10 s) and down (10 s) in the beginning of the session. We used online stimulation (i.e. stimulation during training), as this protocol has been previously shown to enhance the lasting effect of training (*O'Shea et al., 2017*). This facilitatory effect is not present or polarity specific when stimulation precedes training, and both types of stimulation (anodal vs. cathodal) impede learning (*Stagg et al., 2011c*).

To achieve consistent electrode placement across participants when targeting the right posterior occipito-temporal cortex (consistent with the MRS acquisition in the right occipito-temporal cortex), we placed the bottom right corner of the square stimulating electrode on T6, using a 10–20 system EEG cap, maintaining the same orientation across participants, parallel to the line connecting T6 and O2. The reference electrode was placed on Cz. We used FreeSurfer (*Dale et al., 1999*) to reconstruct head models from anatomical scans and SimNIBS 2.0.1 (*Thielscher et al., 2015*) to simulate electric field density resulting from stimulation over the grey matter surface (*Figure 4—figure supplement 1*). This analysis showed that the current density was largely unilaterally localized, the peak of the electric field density was observed under the stimulating electrode around the posterior occipito-temporal cortex and the stimulation reached the occipito-temporal region where the MRS voxel was placed.

## Data analysis

### Behavioral data analysis

We employed a single interval forced choice task, where participants were asked to choose between two stimulus classes (radial or concentric) in each trial. To quantify discriminability between the two Glass patterns classes (radial vs. concentric), we computed d' (*Stanislaw and Todorov, 1999*) across trials per run, as the difference between the z-transform of each stimulus class' hit and false alarm rates. In particular, if the stimulus was radial (tR) and the participant responded 'radial' (rR), this was counted as a hit for the radial class (tRrR) or a correct rejection for the concentric class. If the stimulus was radial (tR) and the participant responded 'concentric' (rC), this was counted as a miss for the radial class (tRrC) or a false alarm for the concentric class. When calculating response rates, we computed hit rate for radial and concentric as follows:

Radial Hit Rate: tRrR/tR, Radial False Alarm Rate: tCrR/tC
Concentric Hit Rate: tCrC/tC, Concentric False Alarm Rate: tRrC/tR

Also:

Radial Hit Rate +Concentric False Alarm Rate = tRrR/tR + tRrC/tR = tR/tR = 1 and

Concentric Hit Rate +Radial False Alarm Rate = tCrC/tC + tCrR/tC = tC/tC = 1

d' can be computed using the Radial or Concentric Hit and False Alarm Rates as shown below:

d'=z (Radial Hit Rate) - z (Radial False Alarm Rate)

=z (1-Concentric False Alarm Rate) - z (1-Concentric Hit Rate)

= -z (Concentric False Alarm Rate)+z (Concentric Hit Rate),

where z is the inverse cumulative distribution function for a normal distribution (0,1).

To quantify behavioral improvement, we calculated: a) learning rate that indicates the rate of change in perceptual sensitivity as measured by d' per training run, b) Δd' that indicates difference in perceptual sensitivity early (first run) vs. late (last run) in training. To compute learning rate, we fitted individual participant training data with a logarithmic function: $y = k * \ln x + c$, where $x$ is the training run, $y$ is the run d', $c$ is the starting d' and $k$ corresponds to the learning rate, using MATLAB 2013a (The MathWorks, Natick, MA, USA). Positive learning rate indicates that performance improved with training, whereas negative or close to zero learning rate indicates no behavioral improvement.

To compare behavioral performance between the different tDCs groups for the two tasks, we run a repeated-measures ANOVA, with training-block and stimulation as factors, using SPSS (IBM Corporation, Armonk, NY, USA). To directly compare the two tasks, we normalized behavioral performance (d') in the active stimulation groups (anodal, cathodal) to the sham stimulation group. For each block, we computed the average d' across participants in the sham group. We subtracted this mean d' per block from each participant's data in the anodal and cathodal groups. We then calculated the group mean d' for each block normalized to sham and conducted a repeated-measures ANOVA on the data from the active stimulation groups (anodal, cathodal) normalized to the sham group, with task, stimulation and training-block as factors. We used Greenhouse-Geisser (for epsilon less than 0.75) and Huynh-Feldt (for epsilon greater than 0.75) corrections of significance.

## MRS data analysis

We pre-processed MRS data according to (*Prescot and Renshaw, 2013*), using MATLAB 2013a (The MathWorks, Natick, MA, USA) and the prior-knowledge fitting software ProFit (*Schulte and Boesiger, 2006*). We assessed the quality of the fit by means of visual inspection and calculation of the Cramer-Rao Lower Bounds (CRLB) of variance. Only participant data without contamination (e.g. due to lipids) (*Figure 2—figure supplement 2*) and GABA CRLB values < 20% for both pre- and post-training fitted data were included in further steps of MRS related analyses (data from three participants for the Feature-differences task were excluded based on these criteria). Residual water was removed from each row of water suppressed 2D matrices using a Hankel singular value decomposition (HSVD) MATLAB routine (*Cabanes et al., 2001*; *Prescot and Renshaw, 2013*). We referenced metabolite concentrations to the concentration of total Creatine (tCre). tCre has been widely used as a reference metabolite in MRS studies (*Donahue et al., 2010*; *Sampaio-Baptista et al., 2015*) and this referencing method has been shown to have better reproducibility compared to other methods (*Bogner et al., 2010*). We then subtracted pre- from post-training concentrations to estimate GABA/tCre changes before compared to after training.

To account for the variability in tissue composition within the MRS voxel across participants, we calculated the percentage of grey matter (GM), white matter (WM) and cerebrospinal fluid (CSF) in each of the MRS measurement voxels. We conducted whole brain tissue-type segmentation of the T1-weighted anatomical scan using FAST (*Zhang et al., 2001*), in the FMRIB Software Library (*Smith et al., 2004*). We then divided GABA concentration by GM/(GM + WM + CSF) and tCre concentration by (GM + WM)/(GM + WM + CSF) (*Kolasinski et al., 2017*).

We used bootstrapped Pearson's correlations to measure the linear association between variables (GABA, behavioral improvement, BOLD change) as implemented in the Robust Correlation toolbox (*Pernet et al., 2012*). Skipped-correlations detect bivariate outliers and account for their removal when testing for correlation significance. Bivariate outliers were detected using the box-plot rule on z-scored values: the algorithm calculates orthogonal distances of all data points from the center of the bivariate distribution and marks as outliers data points with distances that exceed the interquartile range. Where bivariate outliers were detected we reported Skipped Pearson's r and

bootstrapped confidence intervals. Note that bivariate outliers are not shown in the data figures. We used Fisher's test to compare correlation metrics between tasks and Steiger's test within task.

## fMRI data pre-processing

We used BrainVoyager QX 2.8 (Brain Innovation, Maastricht, The Netherlands) for fMRI data analysis (*Goebel et al., 2006*). We used an automated alignment routine (rigid body transformation) together with manual adjustments to ensure precise co-registration of the functional and anatomical data. T1-weighted anatomical data were used for co-registration, and three-dimensional head motion correction, temporal high-pass filtering (Fast Fourier Transform with a cut-off of 3 cycles) and removal of linear trends. Trials with motion larger than 3 mm were excluded from further analyses. Three participants with excessive motion across experimental runs were excluded from fMRI related analyses (one for the Feature-differences task and two for the Signal-in-Noise task). Spatial smoothing (Gaussian filter, full-width-at-half-maximum 5 mm) was used for group GLM analysis. The functional images were aligned to anatomical data under careful visual inspection, and the complete data were transformed into Talairach space (nearest-neighbor interpolation). The functional runs were co-aligned to the first functional volume of the first run of the session.

## fMRI data analysis

To investigate fMRI learning-dependent changes during training, we analyzed the data using a General Linear Model (GLM) with two task related regressors (stimulus vs. fixation trials) and six head movement regressors based on the motion correction parameters. We conducted a whole-brain voxel-wise covariance analysis to identify voxel clusters that show significant correlations between BOLD activation change (late: last two training runs vs. early: first two training runs) and behavioral improvement (learning rate, $\Delta d'$). To assess the relationship between learning-dependent changes in fMRI and GABA measurements, we conducted a covariance analysis that tested for voxels that showed significant correlation between fMRI activity change (late vs. early training runs) and GABA change (post- minus pre-training GABA measurement). We used Brain Voyager's cluster-extent thresholding tool (ClusterThresh plugin) and ran Monte Carlo simulations to estimate the cluster-extent threshold and confirm a family wise error threshold of p=0.05.

## Acknowledgements

The authors are grateful to Rui Wang, Rachel Bellamy and Timothy Lee for help with the data collection, Adrian Garcia for preliminary work, Andrew Welchman and Jasper Poort for helpful comments, Charlotte Stagg and Holly Bridge for helpful discussions and Andrew Prescot for providing MRS acquisition tools to MRC-CBU, Cambridge.

## Additional information

### Funding

| Funder | Grant reference number | Author |
| --- | --- | --- |
| Biotechnology and Biological Sciences Research Council | BB/P021255/1 | Zoe Kourtzi |
| Wellcome | 205067/Z/16/Z | Zoe Kourtzi |
| Seventh Framework Programme | FP7/2007-2013 | Zoe Kourtzi |
| Alan Turing Institute | TU/B/000095 | Zoe Kourtzi |

The funders had no role in study design, data collection and interpretation, or the decision to submit the work for publication.

### Author contributions

Polytimi Frangou, Conceptualization, Data curation, Software, Formal analysis, Validation, Investigation, Visualization, Methodology, Writing—original draft, Project administration, Writing—review

and editing; Marta Correia, Resources, Data curation, Software, Investigation, Methodology, Writing—review and editing; Zoe Kourtzi, Conceptualization, Resources, Supervision, Funding acquisition, Investigation, Methodology, Writing—original draft, Project administration, Writing—review and editing

### Author ORCIDs
Polytimi Frangou (iD) http://orcid.org/0000-0003-3524-0306
Marta Correia (iD) http://orcid.org/0000-0002-3231-7040
Zoe Kourtzi (iD) http://orcid.org/0000-0001-9441-7832

### Ethics
Human subjects: Participants gave written informed consent. The study was approved by the University of Cambridge ethics committee (reference number CPREC.PRE.2013.147).

### Decision letter and Author response
Decision letter https://doi.org/10.7554/eLife.35854.026
Author response https://doi.org/10.7554/eLife.35854.027

## Additional files
### Supplementary files
• Transparent reporting form
DOI: https://doi.org/10.7554/eLife.35854.022

### Data availability
Source data files have been provided for Figures 1,2,3,4. Data can also be found on the Cambridge Data Repository: https://doi.org/10.17863/CAM.30241

The following dataset was generated:

| Author(s) | Year | Dataset title | Dataset URL | Database, license, and accessibility information |
|---|---|---|---|---|
| Frangou P, Correia M, Kourtzi Z | 2018 | Data supporting 'GABA, not BOLD, reveals dissociable learning-dependent plasticity mechanisms in the human brain' | https://doi.org/10.17863/CAM.30241 | Publicly available Apollo - University of Cambridge Repository |

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
