## [Decision Letter]

Thank you for submitting your article "GABA – not BOLD – reveals dissociable learning mechanisms in the human visual cortex" for consideration by *eLife*. Your article has been reviewed by three peer reviewers, including Michael Silver as a guest Reviewing Editor and Reviewer #3, and the evaluation has been overseen by David Van Essen as the Senior Editor. The following individuals involved in review of your submission have agreed to reveal their identity: Barbara Anne Dosher (Reviewer #1) and Ian Greenhouse (Reviewer #2).

The reviewers have discussed the reviews with one another and the Reviewing Editor has drafted this decision to help you prepare a revised submission.

Summary:

The reviewers agree that your discovery of a double dissociation of changes in visual cortical GABA levels and behavioral measures of learning following training in two different tasks (extracting signal from noise and feature discrimination) is novel and important. Also, the use of tDCS to manipulate cortical excitability to test the hypotheses generated by your initial findings is a strength of your study, and your tDCS results provide strong support for the validity of your predictions. The reviewers are also impressed by the use of multiple measurements to provide converging evidence for your conclusions.

However, there are a number of outstanding issues in your submission that should be addressed. First, there are concerns about the fact that different measures of behavioral performance (proportion correct and d') were used to compute learning rate and learning magnitude, particularly since you report a surprising double dissociation of the correlations of these behavioral measures with changes in GABA for the two tasks. It's also not clear how you computed d'. More explanation and justification is needed for these procedures for calculating the behavioral correlates of learning, and you should provide more interpretation of the double dissociation you observed.

The reviewers raised additional concerns regarding your decision to remove the 8th training run from your analysis without sufficient justification. In addition, you should report overall effects of training on GABA levels and BOLD responses in your data, thereby allowing you to assess if you were able to replicate previous reports from the literature with respect to these training-induced changes. Also, more description is needed regarding the repeated measures ANOVAs and which statistical comparisons were made between groups versus within group. Finally, you should address to what extent changes in the scanner environment could have accounted for your observed changes in BOLD responses and GABA levels.

Essential revisions:

1) The authors measure behavioral effects of learning in two ways: change in d' and learning rate (derived from fitting a logarithmic function to accuracy versus time plots). The first measure quantifies the magnitude of learning, while the second quantifies the rate of learning. For the Feature-differences task, changes in GABA were positively correlated with d' change but were not significantly correlated with learning rate. For the Signal-in-noise task, changes in GABA were negatively correlated with learning rate but not significantly correlated with d' change.

This double dissociation is unexpected. Why would GABA track only a measure of the magnitude of learning (but not its rate) in one task and the rate of learning (but not its magnitude) in the other? The authors need to provide more discussion on this point. Also, the authors write "…GABA change related to enhanced performance – as measured by learning rate – when training on target detection from clutter, while enhanced stimulus discriminability – as measured by d' – when training on fine feature discrimination." Learning rate is not the same as "enhanced performance" – it is a rate, not a magnitude, measure. In fact, d' change quantifies the amount of enhancement in performance. Additionally, enhanced stimulus discriminability is quantified as a change in d', not d' itself.

In the third paragraph of the subsection “Relating GABA to behavioral improvement and BOLD change”, the authors describe differences in the component processes that are engaged by the two tasks. However, the GABA/behavior correlations are based on learning of the tasks, not task performance per se.

2) Why is behavioral performance expressed as accuracy (proportion correct) in some cases (e.g., Figure 1B, Figure, 4, calculation of learning rate) and d' in others (e.g., Figure 1C)? In particular, it is surprising that learning rate is not estimated from d' values, given that the other behavioral measure (d' change) is obviously based on d' values. Also, how is d' defined in this 2-alternative forced choice task? If the stimulus is radial and the subject responds "radial", is this considered to be a hit for radial or a correct rejection for concentric? If the stimulus is radial and the subject responds "concentric", is this a false alarm for concentric or a miss for radial?

3) The authors removed the 8th training run from their analysis and state that this was done due to performance decline possibly resulting from fatigue. This data exclusion should be justified by providing more information about the decline in performance. Also, why might fatigue be expected to impact behavior and BOLD measurements but not GABA measurements, which were obtained after the 8th training run? Would the correlations between performance measures and GABA change and between BOLD change and GABA change be stronger if the imaging study had stopped after 7 training runs?

4) Previous studies are cited that showed changes in GABA content that resulted from training and stimulation interventions. It would be helpful to know if this study observed similar changes in the primary measures of interest as a result of training. Were there significant post-pre training differences in GABA measurements at the group level? Similarly, were there significant post-pre training differences in BOLD activations at the group level? Were changes in GABA and/or BOLD consistent between the two task groups (SN vs. FD)? This information would help relate the current results to previous studies to better understand the correlations within groups.

In addition, if changes in BOLD are similar in the SN and FD tasks (as appears to be the case in Figure 3B), why are these changes associated with changes in GABA in opposite directions in the two tasks? This is written as though only GABA measures could discriminate the two mechanisms, but other test designs, perhaps ones that manipulate external noise and feature differences, might lead to clear differences in fMRI signals.

5) The differing correlations between GABA and SN task learning rate and between GABA and FD task d' could reflect group differences. The Results section should clearly indicate that two different groups were run in the SN and FD tasks. The language is somewhat misleading in this section, as it seems to imply a within-subjects design, when in fact this was not the case. A clear description should be provided regarding which statistical comparisons were between groups and which were within groups as well as how the different groups were modeled in the repeated measures ANOVAs.

6) Correlations between changes in BOLD and changes in MRS measurements could reflect underlying changes in the scanner environment that impact both measures. Did the authors account for B0 field changes that can arise due to head motion and/or gradient heating and cooling over extended blocks of EPI sequences?

7) What do a negative learning rate and a negative change in d' represent? If participants get worse at the task with training, this suggests that there may be additional mechanisms that influenced the observed correlations other than learning.

---

## [Author Response]

Essential revisions:1) The authors measure behavioral effects of learning in two ways: change in d' and learning rate (derived from fitting a logarithmic function to accuracy versus time plots). The first measure quantifies the magnitude of learning, while the second quantifies the rate of learning. For the Feature-differences task, changes in GABA were positively correlated with d' change but were not significantly correlated with learning rate. For the Signal-in-noise task, changes in GABA were negatively correlated with learning rate but not significantly correlated with d' change.This double dissociation is unexpected. Why would GABA track only a measure of the magnitude of learning (but not its rate) in one task and the rate of learning (but not its magnitude) in the other? The authors need to provide more discussion on this point.

Following, the reviewers’ suggestion, we calculated learning rate on d’ – rather than accuracy – data (see also our first response to point 2). These analyses showed the same pattern of results, as previously reported; that is, learning rate (but not Δd’) correlates negatively with GABA change for the Signal-in-Noise task, while Δd’ (but not learning rate) correlates positively with GABA change for the Feature differences task. We have now corroborated this result across different analyses: a) relating behavior to GABA across all participants vs. including data only from participants with positive learning rate or Δd’, b) controlling for baseline GABA concentration, tissue composition within the MRS voxel (GM,WM,CSF), and changes in glutamate. We have now included these analyses on d’ data in the revised manuscript and figures.

This double dissociation suggests that the relationship between GABAergic plasticity (as indicated by changes in GABA concentration with training) and behavioral improvement is task-dependent. We further test this hypothesis in a moderation analysis that shows that task significantly moderates the relationship of GABA change with a) behavioral improvement, and b) BOLD change with training. We agree with the reviewers that this double dissociation is intriguing. Interestingly, recent studies characterizing the role of different populations of interneurons in visual learning may shed light into this task-dependent GABAergic plasticity. In particular somatostatin-positive (SOM) interneurons have been implicated in spatial summation (Adesnik et al., 2012) and have been shown to act during training to gate plasticity by providing contextual information (van Versendaal and Levelt, 2016). In contrast, parvalbumin-positive (PV) interneurons have been implicated in selective inhibition (Rokem et al., 2011) that sharpens feature representations after training (Khan et al., 2018). It is therefore possible that the dissociable correlations we observed between tasks for GABA change and behavioral improvement may reflect differential involvement of SOM vs. PV interneurons in the two tasks. Specifically, SOM interneurons involved in spatial integration may support learning to detect targets from clutter (SN task) through noise filtering. In contrast, PV interneurons involved in selective inhibition may support learning fine differences (FD task) through re-tuning of feature templates.

Further, SOM vs. PV interneurons are shown to be involved in different stages during the time course of learning. In particular, SOM cells have been shown to gate learning-dependent plasticity *during training* (Chen et al., 2015), while PV cells form stimulus-specific ensembles with pyramidal cells *after training* on a visual discrimination task (Khan et al., 2018). Thus, it is possible that different behavioral measures capture the function of SOM vs. PV interneurons, consistent with the double dissociation we observed between tasks for the correlations of GABA change and behavioral improvement. In particular, learning rate (i.e. the rate with which perceptual sensitivity changes during training) may capture best the function of SOM interneurons that act during learning to support noise filtering throughout the course of training. In contrast, Δd’ (i.e. change in perceptual sensitivity after training) may capture best the function of PV interneurons that are shown to support tuning of stimulus-specific representations after training.

In sum, the double dissociation we observed reveals task-dependent GABAergic plasticity mechanisms and proposes testable hypotheses about the micro-circuits (i.e. SOM vs. PV interneurons) that give rise to learning by noise filtering vs. feature template retuning. Further animal studies are needed to test these hypotheses and resolve the detailed microcircuits that underlie the macroscopic learning-dependent plasticity as measured by human brain imaging.

Also, the authors write "…GABA change related to enhanced performance – as measured by learning rate – when training on target detection from clutter, while enhanced stimulus discriminability – as measured by d' – when training on fine feature discrimination." Learning rate is not the same as "enhanced performance" – it is a rate, not a magnitude, measure. In fact, d' change quantifies the amount of enhancement in performance. Additionally, enhanced stimulus discriminability is quantified as a change in d', not d' itself.

We thank the reviewers for pointing this out. We have now clarified that a) Δd’ indicates difference in perceptual sensitivity early vs. late in training, b) learning rate indicates the rate with which perceptual sensitivity changes during training.

In the third paragraph of the subsection “Relating GABA to behavioral improvement and BOLD change”, the authors describe differences in the component processes that are engaged by the two tasks. However, the GABA/behavior correlations are based on learning of the tasks, not task performance per se.

We thank the reviewers for this clarification. We have now revised the text and clarified that our results provide evidence for the link between learning (i.e. behavioral improvement) and GABAergic changes in occipito-temporal cortex. Our findings suggest that the component processes involved in each task (i.e. noise filtering vs. feature template returning) relate to GABAergic processing that is altered by learning.

2) Why is behavioral performance expressed as accuracy (proportion correct) in some cases (e.g., Figure 1B, Figure, 4, calculation of learning rate) and d' in others (e.g., Figure 1C)? In particular, it is surprising that learning rate is not estimated from d' values, given that the other behavioral measure (d' change) is obviously based on d' values.

Following the reviewers suggestion, we have now calculated d’ – rather than accuracy (i.e. percent correct) – for both the SN and FD tasks and estimated two measures of behavioral improvement: a) Δd’ that indicates differences in sensitivity with training, b) learning rate that indicates the rate of sensitivity change with training. We have now revised all analyses and figures (Figure 1, 2, 3, 4) to report calculations based on d’ rather than accuracy data.

In particular, we revised the analyses in Figure 1 as follows. First, we found no significant difference in learning rate calculated on d’ data between the two tasks (t(34)= 0.033, p=0.974), similar to the analysis we previously reported based on accuracy (t(34)= 0.584, p=0.563) (Figure 1a). Second, similar areas were activated in the covariance analysis of BOLD change with behavioral improvement for learning rate calculated on d’ data, as with the accuracy learning rate measure that was used previously (Figure 1B, Figure 1—source data 1).

For the analyses in Figure 2, we found a significant negative correlation between GABA change and learning rate calculated on d’ data (r=-0.43, CI=[-0.74, -0.07]) for the Signal in Noise task (as previously reported for accuracy learning rate: r=-0.48, CI=[-0.74, -0.15]) (Figure 2). This correlation was significantly different from a significant positive correlation between GABA change and Δd’ for the Feature differences task (Fisher’s Z=2.91, p=0.004). We found no significant correlation between GABA change learning rate for the Feature differences task (r= 0.13, CI=[-0.38, 0.62]).

For Figure 3, we corroborated the result of the covariance analysis of fMRI with GABA change by extracting BOLD signal from the voxel clusters in the posterior occipito-temporal cortex that resulted from the revised covariance analysis of fMRI with behavioral improvement (Figure 1B). Correlations of change in GABA and BOLD – extracted from this independently defined region of interest – were significant (SN: r=-0.58, CI=[-0.83, -0.20]; FD: r=0.70 CI=[0.40 0.89]) and significantly different (Z= 4.19, p=0.00003) between the two tasks (Figure 3B).

For analyses in Figure 4, showing the effect of tDCs on behavioral performance, we computed d’ instead of accuracy. The results remained as before, as indicated by the statistical analyses reported in the text and the revised Figure 4. In particular, we found significant behavioral improvement for anodal compared to sham stimulation for the Signal-in-Noise (Training block x Stimulation: F(2,52.5)=3.99, p=0.02), but not the Feature differences task (Training block x Stimulation: F(2.2,57.9)=0.45, p=0.66). In contrast, we observed improved performance during cathodal compared to sham tDCs for the Feature-differences task (main effect of stimulation: F(1,26)=6.13, p=0.02), but not the Signal-in-Noise task (main effect of stimulation: F(1,26)=0.001, p=0.98). To compare behavioral improvement between tasks, we normalized performance during tDCs (anodal or cathodal) to performance during sham stimulation (Figure 4A). A repeated-measures ANOVA showed a significant Task, Stimulation x Training block interaction F(2.5,130)= 3.193, p=0.03).

Also, how is d' defined in this 2-alternative forced choice task? If the stimulus is radial and the subject responds "radial", is this considered to be a hit for radial or a correct rejection for concentric? If the stimulus is radial and the subject responds "concentric", is this a false alarm for concentric or a miss for radial?

We thank the reviewers for this clarification and have revised the text in the Materials and methods section (subsection “Behavioral data analysis”). In particular, we employed a single interval forced choice task, where participants were asked to choose between two stimulus classes (radial or concentric) in each trial. To quantify discriminability between the two Glass patterns classes (radial vs. concentric), we computed d’ (Stanislaw and Todorov, 1999) across trials per run, as the difference between the z-transform of each stimulus class’ hit and false alarm rates. In particular, if the stimulus was radial (tR) and the participant responded “radial” (rR), this was counted as a hit for the radial class (tRrR) or a correct rejection for the concentric class. If the stimulus was radial (tR) and the participant responded "concentric” (rC), this was counted as a miss for the radial class (tRrC) or a false alarm for the concentric class.

Trial Radial (tR)Trial Concentric (tC)ResponseRadial (rR)rRtRrRtCResponseConcentric (rC)rCtRrCtC

When calculating response rates, we computed hit rate for radial and concentric as follows:

Radial Hit Rate: tRrR /tR, Radial False Alarm Rate: tCrR/tC

Concentric Hit Rate: tCrC/tC, Concentric False Alarm Rate: tRrC/tR

Also:

Radial Hit Rate + Concentric False Alarm Rate = tRrR/tR + tRrC/tR = tR/tR = 1

and

Concentric Hit Rate + Radial False Alarm Rate = tCrC/tC + tCrR/tC = tC/tC = 1

d’ can be computed using the Radial or Concentric Hit and False Alarm Rates as shown below:

d' = z (Radial Hit Rate) – z (Radial False Alarm Rate)

= z (1-Concentric False Alarm Rate) – z (1-Concentric Hit Rate)

= -z (Concentric False Alarm Rate) + z (Concentric Hit Rate),

where z is the inverse cumulative distribution function for a normal distribution (0,1).

3) The authors removed the 8th training run from their analysis and state that this was done due to performance decline possibly resulting from fatigue. This data exclusion should be justified by providing more information about the decline in performance. Also, why might fatigue be expected to impact behavior and BOLD measurements but not GABA measurements, which were obtained after the 8th training run? Would the correlations between performance measures and GABA change and between BOLD change and GABA change be stronger if the imaging study had stopped after 7 training runs?

We excluded the 8^th^ run from the analyses, as we were missing data from this run for several participants (n=9). As scanning time for data collection was limited, it was not possible to complete the 8^th^ run for all participants within the time allocated for the fMRI scanning and before continuing to the 2^nd^ GABA measurement after training. Below we present the results of our analyses including the 8th training run, when available. The results are highly similar to those presented in the paper (where 7 experimental runs were used). We clarify this in the revised text and apologize for the confusion that our previous description may have caused by referring to fatigue. It is clear that participants continued to improve in their performance (Figure 1—figure supplement 4A), therefore we have no good evidence for fatigue. It is not possible to estimate what the effect of an additional training run (i.e. 5 minutes 45 seconds of training) may have been on GABA concentration, as the temporal resolution of our MRS GABA measurements on the 3T is much longer (i.e. 17 minutes for 1 GABA measurement) than this timescale.

In the revised manuscript, we chose to present data from all participants for the same training duration (i.e. including 7 runs). This ensured that our statistical analyses were not affected by missing data for 9 participants when including 8 runs. Below, we outline the analyses including participant data with 8 runs and present these results as supplementary figures (Figure 1—figure supplement 4, Figure 2—figure supplement 4B).

First, we show that behavioral improvement is similar when eight vs. seven runs are included in the analyses. Including participant data with 8 runs did not show any significant differences between the two tasks for learning rate (t(34)=0.09, p=0.929) nor Δd’ (t(34)=0.14, p=0.886) (Figure 1—figure supplement 4A). Second, similar areas were activated in the covariance analysis of BOLD change with behavioral improvement (Figure 1—figure supplement 4B) when we included participant data with 8 runs. The results showed positive correlations of BOLD change with learning rate and Δd’ (Figure 1—figure supplement 4B, C). Third, correlations of GABA change with behavioral improvement remained significantly different between tasks (Fisher’s z= 2.4, p=0.02) when we included participant data with 8 runs (Figure 2—figure supplement 4B). Finally, similar areas were activated for the covariance analysis of fMRI with GABA change when the 8th run was included in the fMRI data (Figure 3—figure supplement 1A). We corroborated this result by extracting BOLD signal from the voxel clusters in the posterior occipito-temporal cortex that resulted from the covariance analysis of fMRI with behavioral improvement (Figure 1—figure supplement 4B). Correlations of change in GABA and BOLD – extracted from this independently defined region of interest were significantly different (Fisher’s z= 3.26, p=0.001) between the two tasks (Figure 3—figure supplement 1B).

4) Previous studies are cited that showed changes in GABA content that resulted from training and stimulation interventions. It would be helpful to know if this study observed similar changes in the primary measures of interest as a result of training. Were there significant post-pre training differences in GABA measurements at the group level? Similarly, were there significant post-pre training differences in BOLD activations at the group level? Were changes in GABA and/or BOLD consistent between the two task groups (SN vs. FD)? This information would help relate the current results to previous studies to better understand the correlations within groups.

We thank the reviewers for this suggestion and have revised the manuscript to include further analyses and discussion on this point.

In particular, we observed changes in mean fMRI activation in the occipito-temporal cortex with training (i.e. BOLD comparison across runs) for both tasks (Figure 1—figure supplement 3), consistent with previous fMRI studies on perceptual learning tasks (Mukai et al., 2007). For more discussion see also our second response to point 4, below.

However, we did not find significant differences in mean GABA concentration before vs. after training (Figure 2—figure supplement 3) for either task (main effect of MRS block: F(1,34)= 0.06, p=0.81; Task x MRS block interaction: F(1,34)= 0.21, p=0.65). Only a handful of previous studies have reported mean changes in GABA concentration in the motor and visual cortex due to training. Specifically, GABA in the motor cortex has been shown to decrease significantly after 6 weeks of motor training (juggling practice) (Sampaio-Baptista et al., 2015) and decline steadily during training on a motor task (Floyer-Lea et al., 2006). Further, significantly decreased GABA was found in the primary visual cortex after 150min of monocular deprivation (Lunghi et al., 2015). Note that a recent visual learning study showed differences in the ratio of excitation/inhibition – but, no significant changes in GABA alone – after participant performance plateaued due to training (Shibata et al., 2017).

The main difference between our study and these previous reports is that participant performance increased but did not saturate during the single training session employed in our study (i.e. participant reached mean performance 74%), in contrast to previous studies that showed saturated performance after training. Our previous studies (Li, Mayhew, and Kourtzi, 2012; Mayhew, Li, and Kourtzi, 2012) using the same training paradigm and stimuli have shown that performance saturated after multiple training sessions rather than a single training session, as adopted in our study. Thus, it is likely that mean changes in GABA concentration are more pronounced when participant performance has plateaued after training. Further, it is likely that 7T imaging (rather than 3T imaging used in our study) affords increased signal-to-noise ratio and time resolution that may benefit measurement of change in GABA concentration (Barron et al., 2016; Lunghi et al., 2015).

In addition, if changes in BOLD are similar in the SN and FD tasks (as appears to be the case in Figure 3B), why are these changes associated with changes in GABA in opposite directions in the two tasks? This is written as though only GABA measures could discriminate the two mechanisms, but other test designs, perhaps ones that manipulate external noise and feature differences, might lead to clear differences in fMRI signals.

Our fMRI results showed similar learning-dependent changes in BOLD for both tasks, suggesting that BOLD changes at this early stage of learning (i.e. single training session that resulted in maximum 74% mean performance) do not differ between tasks. This is consistent with a previous fMRI study that showed learning-dependent changes within a single training session (Mukai et al., 2007).

We have now included further discussion of this point in the revised manuscript (subsection “Learning-dependent changes in behavior and fMRI”, last paragraph).

Yet, as we discuss in the manuscript, BOLD reflects the aggregate activity of excitatory and inhibitory signals at the scale of large neural populations and therefore does not allow us to discern the contribution of inhibitory mechanisms to learning. Investigating how learning-dependent changes in BOLD and GABA change relate revealed dissociable mechanisms between the two tasks. In particular, we observed negative correlations of BOLD and GABA change for the signal-in-noise task, suggesting that learning to detect targets in clutter is implemented by decreased local suppression that facilitates recurrent processing for noise filtering and target detection. In contrast, we observed positive correlations of BOLD and GABA change for the fine discrimination, suggesting that learning to discriminate feature differences is implemented by increased GABA that may relate to enhanced feature selectivity and facilitate fine visual discriminations.

We have now included further discussion of this point in the revised manuscript (Discussion, first paragraph).

It is possible that the two tasks may show discriminable BOLD activations after more extensive training that results in saturated behavioral performance. Our behavioral results showed that participant performance did not saturate within the single training session implemented in our study. In our earlier work (Kourtzi et al., 2005; Li et al., 2012; Mayhew et al., 2012) using similar learning paradigms comprising multiple training sessions, we have shown dissociable changes in BOLD between signal-in-noise vs. fine discrimination tasks. Similarly, paradigms that manipulate task difficulty by varying the noise or stimulus similarity may result in different BOLD signatures between tasks.

We have now included further discussion of this point in the revised manuscript (subsection “Learning-dependent changes in behavior and fMRI”, last paragraph).

5) The differing correlations between GABA and SN task learning rate and between GABA and FD task d' could reflect group differences. The Results section should clearly indicate that two different groups were run in the SN and FD tasks. The language is somewhat misleading in this section, as it seems to imply a within-subjects design, when in fact this was not the case. A clear description should be provided regarding which statistical comparisons were between groups and which were within groups as well as how the different groups were modeled in the repeated measures ANOVAs.

We have revised the text to clearly explain that we trained 2 separate groups of participants; one group on the SN task and another on the FD task. Each group was trained only on one of the two tasks (SN, FD) to avoid transfer effects across tasks that have been previously reported when the same individuals were trained sequentially on both tasks (Chang, Kourtzi, and Welchman, 2013; Dosher and Lu, 2007).

To test whether the dissociable relationships between GABA change and behavioral improvement for the two tasks could be due to differences across participants per group, we compared the behavioral and imaging data before training. First, we did not find any significant differences in GABA concentration (t(34)=0.11, p=0.91) nor in behavioral performance (t(34)=0.23, p=0.82) between the two groups before training. Second, we compared signal-to-noise ratio (SNR) across tasks for the first MRS measurement (i.e. pre-training) and the first two EPI runs (i.e. early in the training, as there were no EPI measurements before training). We did not find any significant differences in MRS SNR (t(34)=0.77, p=0.45), nor fMRI temporal SNR (tSNR) between the two tasks (t(34)=0.73, p=0.47). These control analyses suggest that the dissociable results we observed between tasks could not be simply due to individual differences across participants that were trained on different tasks. We discuss this point and include these analyses in the revised manuscript (subsection “Control analyses”, first paragraph).

6) Correlations between changes in BOLD and changes in MRS measurements could reflect underlying changes in the scanner environment that impact both measures. Did the authors account for B0 field changes that can arise due to head motion and/or gradient heating and cooling over extended blocks of EPI sequences?

We conducted the control analyses described below, to test for differences in the scanner environment across training and tasks. Our results suggest that it is unlikely that the dissociable correlations between BOLD and GABA we observed between tasks could be due to differences in the scanner environment. We explain these analyses below and in the revised manuscript (subsection “Control analyses”, second paragraph).

First, we calculated the variation of the scanner center frequency (SCF) across EPI runs for each participant. Proper spatial localization requires that the SCF is correctly set and stable during the session. We found that the mean SCF variation across participants was very small (0.0000125 ± 0.0000019 MHz), and there was no significant interaction between EPI run and task (F(1,34)=0.68, p=0.42) that could account for the differences we observed in the correlations between BOLD and GABA between tasks. Further, the EPI data was corrected for head movement and any data with head movement larger than 3mm was removed from further analysis.

Second, we calculated tSNR for the EPI data. We found no significant interaction between EPI run and task (F(1,34)= 1.62, p=0.21) that could account for the differences we observed in the correlations between BOLD and GABA between tasks.

Finally, to control for measurement differences in the MRS before vs. after training we conducted the following analyses. First, to assess measurement quality we calculated spectral SNR for each MRS measurement. This analysis showed no significant interaction between MRS block and task (F(1,34) = 2.37, p=0.13) or main effect of block (F(1,34)= 1.60, p= 0.22). Second, to assess spectral resolution before vs. after training, we calculated peak linewidth for each MRS measurement. This analysis showed no significant interaction between MRS block and task (F(1,34)=0.90, p=0.35) nor a significant main effect of block (F(1,34)= 2.97, p=0.09). These results suggest that the MRS data quality was similar before and after training for both tasks. Thus, differences in the correlations between BOLD and GABA between tasks could not be due to difference in the quality of the GABA measurements.

7) What do a negative learning rate and a negative change in d' represent? If participants get worse at the task with training, this suggests that there may be additional mechanisms that influenced the observed correlations other than learning.

Negative learning rate or negative Δd’ represent decreased sensitivity during training. As participants were trained only for a single training session and without trial-by-trial feedback, these measures may be noisier and result in negative values compared to when they are calculated across multiple training sessions that typically result in increased and saturated performance. To control for the possibility that measurement noise may affect our results, we conducted correlations of GABA and BOLD change with behavioral improvement after removing the data from participants with negative learning rate or Δd’. These analyses showed that the correlations remained significant, suggesting that it is unlikely that our results were confounded by negative values reflecting weaker learning. We described these analyses below and in the revised manuscript (subsection “Control analyses”, last paragraph).

First, when we removed data from participants with negative learning rate (n=3) the correlation of GABA change with learning rate (originally: r=-0.43, CI=[-0.74, -0.07]) remained significant for the Signal-in-Noise task (r=-0.52, CI=[-0.80, -0.09]). Similarly, when we removed data from participants with negative Δd’ (n=8) the correlation of GABA change with Δd’ (originally: r=0.54, CI=[0.05, 0.85]) remained significant for the Feature differences task (r= 0.72, CI=[0.29, 0.94]).

Second, when we removed data from participants with negative learning rate (n=9), BOLD change (early vs late experimental runs) correlated significantly with learning rate (r=0.58, CI=[0.30, 0.77]). Similarly, when we removed data from participants with negative Δd’ (n=12), BOLD change (early vs. late experimental runs) correlated significantly with Δd’ (r=0.42, CI=[0.05, 0.67]).

Finally, we correlated change in BOLD signal that was extracted from the voxel clusters in the posterior occipito-temporal cortex (Figure 1C) with GABA change. The correlations of BOLD change (early vs. late experimental runs) and GABA change (before vs. after training) remained significantly different between the two tasks (Z= 2.84, p= 0.01) after removing data from participants with negative learning rate or Δd’.